



# Dam type and topological position govern ice-marginal lake area change in Alaska and NW Canada between 1984 and 2019

Brianna Rick[1], Daniel McGrath[1], William Armstrong[2], Scott W. McCoy[3]

[1]Department of Geosciences, Colorado State University, Fort Collins, CO 80523, USA
5  [2]Department of Geological and Environmental Sciences, Appalachian State University, Boone, NC 28607, USA
[3]Department of Geological Sciences and Engineering, University of Nevada, Reno, NV 89557, USA

*Correspondence to*: Brianna Rick (brianna.rick@colostate.edu)



**Abstract.** Ice-marginal lakes impact glacier mass balance, water resources, and ecosystem dynamics, and can produce catastrophic glacial lake outburst floods (GLOFs) via sudden drainage. Multitemporal inventories of ice-marginal lakes are a critical first step in understanding the drivers of historic change, predicting future lake evolution, and assessing GLOF hazards. Here, we use Landsat-era satellite imagery and supervised classification to semi-automatically delineate lake outlines for four ~5 year time periods between 1984 and 2019 in Alaska and northwest Canada. Overall, ice-marginal lakes in the region have grown in total number (+176 lakes, 36% increase) and area (+467 km$^2$, 57% increase) between the time periods of 1984–1988 and 2016–2019. However, changes in lake numbers and area were notably unsteady and nonuniform. We demonstrate that lake area changes are connected to dam type (moraine, bedrock, ice, or supraglacial) and topological position (proglacial, detached, unconnected, ice, or supraglacial), with important differences in lake behavior between the sub-groups. In strong contrast to all other dam types, ice-dammed lakes decreased in number (–9, 13% decrease) and area (–56 km$^2$, 43% decrease), while moraine-dammed lakes increased (+59, 28% and +468 km$^2$, 85% for number and area, respectively), a faster rate than the average when considering all dam types together. Proglacial lakes experienced the largest area changes and rate of change out of any topological position throughout the period of study. By tracking individual lakes through time and categorizing lakes by dam type, subregion, and topological position, we are able to parse trends that would otherwise be aliased if these characteristics were not considered. This work highlights the importance of such lake characterization when performing ice-marginal lake inventories, and provides insight into the physical processes driving recent ice-marginal lake evolution.

# 1 Introduction

## 1.1 Ice-marginal lakes

Ice-marginal lakes are located adjacent to glaciers, commonly forming at glacier termini, tributary junctions, along glacier margins, or where glacially eroded bedrock or sediment creates topographic depressions (overdeepenings; Carrivick and Tweed, 2013; Cook and Quincey, 2015). These lakes can impact human societies in a multitude of ways, ranging from water resources (Immerzeel et al., 2020) and tourist attractions (Wang and Zhou, 2019; Welling et al., 2020) to destructive and lethal hazards (Carrivick and Tweed, 2016; Cook et al., 2016; Emmer, 2017). Formation or drainage of lakes can impact ecosystem dynamics by providing or removing a source of stored freshwater, altering the sediment flux within a basin, creating new habitat (e.g., Milner et al., 2008), or altering downstream flow characteristics (Tweed and Carrivick, 2015; Jacquet et al., 2017). Glacial lake outburst floods (GLOFs; when a lake dam suddenly fails or is breached; Clague and Evans, 2000) from ice-marginal lakes can have massive impacts on downstream river channel morphology (e.g., Jacquet et al., 2017), disrupt ecosystems (e.g., Meerhoff et al., 2018), destroy infrastructure, and cause the loss of human lives (e.g., Carrivick and Tweed, 2016). Time-varying inventories of ice-marginal lakes are a critical first step in predicting future lake evolution and assessing GLOF hazards.

Global (Shugar et al., 2020) and regional (Wang et al., 2013; Carrivick and Quincey, 2014; Nie et al., 2017; Song et al., 2017; Emmer et al., 2020) lake inventories document increases in ice-marginal lakes, however, many studies report solely on the



change in total frequency and cumulative area of lakes within the study region, or consider only one type of lake (e.g., moraine-dammed or ice-dammed). Disregarding individual lake characteristics and individual lake changes could obscure important trends and dynamics. If a subset of lakes grow while others drain, reporting the overall lake frequency and area change underestimates total regional changes, as positive and negative growth offset one another. Alternatively, if a few large lakes dominate the documented area change for a region, average lake growth would be overestimated, therefore preventing an

accurate physical understanding of this change. In addition, the ecological consequences of lake changes may differ based on where this growth occurs: a change from no lake to a 1 km$^2$ lake could have larger ecological and hazard implications than a large lake (e.g. >10 km$^2$) expanding by a few square kilometers (Dorava and Milner, 2000). Cumulative regional studies provide a first-order estimate of regional changes, however, individual lake change must be analyzed to develop a process-based understanding of area change in order to predict future lake growth and watershed-scale impacts, such as GLOF hazard

potential.

Recent studies parse lakes by characteristics which influence lake behavior, such as dam type and evolution phase (Emmer et al., 2015, 2020), or the spatial relationship to their source glacier (e.g., proglacial, supraglacial, unconnected, detached; Nie et al., 2017; Rounce et al., 2017; Chen et al., 2021). In the Himalaya, proglacial lakes (located at the glacier terminus, in contact with ice) contributed to 83% of the total lake area increase from 1990 to 2015, while only composing 55% of the lake

population (Nie et al., 2017). Proglacial lakes enhance calving due to ice-water contact, resulting in lake expansion through terminus retreat (Carrivick and Tweed, 2013). In this same region, supraglacial lakes (located on debris-covered glacier surfaces) compose 3.5% of the population in 2015 and experienced the greatest percent increase in area (367%) since 1990, though they only constitute 1.2% of total lake area in 2015 (Nie et al., 2017). Proglacial lakes tend to be persistent and contribute the most to regional area growth, whereas supraglacial lakes are small and experience large spatio-temporal variation

in area and location (Nie et al., 2017; Rounce et al., 2017; Chen et al., 2021). In contrast, non-glacier fed and unconnected glacier fed lakes remained fairly stable or experienced small areal expansion (Nie et al., 2017). In the Cordillera Blanca of Peru, many lakes shifted from proglacial to detached (no longer in contact with ice) and the dominant dam type for proglacial lakes shifted from moraine-dammed in 1948 to bedrock-dammed in 2017 (Emmer et al., 2020). This shift in dominant dam type was reflected in recent GLOF-producing lakes as well, as the most recent GLOFs originated from bedrock-dammed lakes

(Emmer et al., 2020). In both the Himalaya and the Cordillera Blanca, dam type and lake position help characterize lake behavior; this variability would be lost if only cumulative changes were reported. These studies support the inclusion of dam type and topological position in comprehensive regional ice-marginal lake studies, suggesting that analyzing all lakes together masks patterns exhibited by each dam type, location, or subregion.

## 1.2 Aims of this study

Alaska's extensive glacier area (86,725 km$^2$; second largest extent outside the ice sheets; Fig. 1; Farinotti et al., 2019; Zemp et al., 2019), range of climates (maritime, transitional, continental, and Arctic; Miller et al., 1999), and varying glaciological characteristics (valleys, cirques, debris, icefields) allow for a unique abundance of ice-dammed, supraglacial, and large




proglacial lakes. The wide variety in ice-marginal lake types, which likely respond differently to climate change (Field et al., 2021), makes this region particularly susceptible to aliasing of lake changes if considered all together. Although Alaska is

sparsely populated, this region has experienced the largest number and highest frequency of recorded GLOFs globally, accounting for 25% of all historical recorded events (Carrivick and Tweed, 2016). As one of the largest, yet understudied, glacierized mountain regions in the world, characterizing Alaska's ice-marginal lakes and their evolution has important implications for local hazards (e.g., remote roads, the Trans Alaskan Pipeline), ecosystem dynamics, and comparable environments around the world.


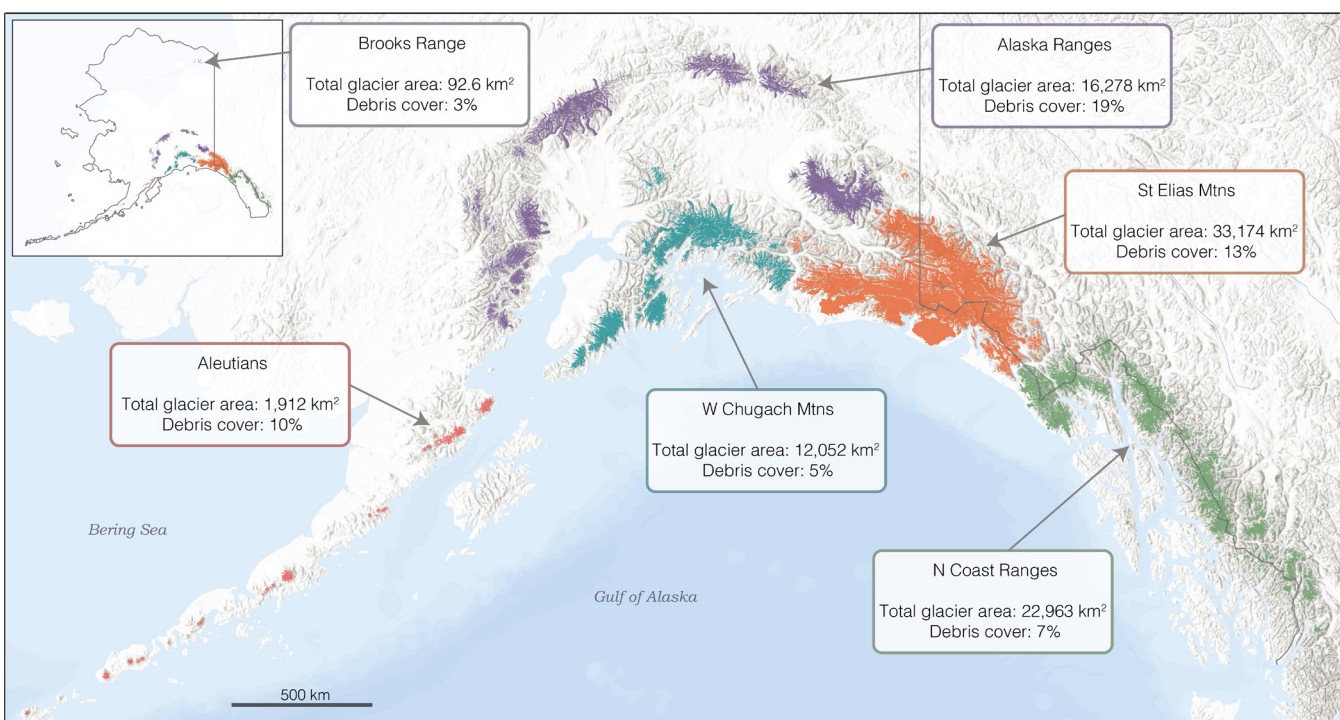

**Figure 1: Distribution of glaciers as mapped by the Randolph Glacier Inventory (RGI Consortium, 2017) and subregions used to separate glaciated areas in Alaska and NW Canada (basemap provided by ESRI, 2021). Total glacier area (km²) and percent debris cover (percent of glacier surface covered by debris; Herreid and Pellicciotti, 2020) are displayed for each subregion.**

Previous studies in Alaska looked specifically at ice-dammed lakes (Post and Mayo, 1971; Wolfe et al., 2014), a subset of lakes (Field et al., 2021), individual case studies (e.g., Sturm and Benson, 1985; Anderson et al., 2003; Kienholz et al., 2020), or a regional subset within a global study (Shugar et al., 2020), none of which comprehensively characterize regional trends. This is the first study to systematically inventory and characterize all ice-marginal lakes in Alaska using Landsat-era satellite imagery between 1984 and 2019. We aim to: (i) characterize ice-marginal lakes and lake area change in Alaska; (ii) determine

whether factors such as dam type, topological position, and region can predict sign of area change; and (iii) explore the



interplay between the spatial distribution of lakes and regional characteristics such as glacier area, glacier complexity, and debris cover.

## 2 Data and methods

### 2.1 Imagery and datasets

Cloud-free mosaics were compiled in Google Earth Engine (GEE), an open source, web-based remote sensing platform. Landsat 5 Thematic Mapper (TM), Landsat 7 Thematic Mapper Plus (ETM+), and Landsat 8 Operational Land Imager (OLI) surface reflectance tier 1 images were used to create 5-year composites (Table 1). Intervals were selected based on available imagery and to capture average lake outlines for a five year period within each decade (1980s to 2010s). Few images are available for the late 1980s to early 1990s, and therefore this time period was excluded. Imagery was limited to a 10 km buffer

around the Randolph Glacier Inventory (RGI v6.0, Region 01; RGI Consortium, 2017; Wang et al., 2012; Zhang et al., 2018). For each pixel within the region of interest, the mosaicking algorithm calculates the median value of all cloud-free pixels between July 1st and September 31st for each year within the given time period (Fig. S1). Five-year mosaics minimize the impact of frequent cloud cover in Alaska, which complicates region-wide assessments on shorter time-scales. In theory, 5-year composites allow for an average shoreline delineation, capturing longer term trends rather than seasonal variation. However,

ice-dammed lakes are known to fill and drain on sub-annual (e.g., Lago Cachet Dos; Jacquet et al., 2017) to annual (e.g., Hidden Creek Lake; Anderson et al., 2003) timescales, so a lake may be characterized as persistent (present at every time step) even though it has undergone multiple fill/drain cycles between composites.

**Table 1: Dataset and the resolution of the dataset used for each time step.**

| Time interval | Dataset | Sensor | Resolution | # of images used |
|---|---|---|---|---|
| 1984–1988 | Landsat 5 | ETM | 30 m | 683 |
| 1997–2001 | Landsat 5 & 7 | ETM/ETM+ | 30 m | 1312 |
| 2007–2011 | Landsat 5 & 7 | ETM/ETM+ | 30 m | 2630 |
| 2016–2019 | Landsat 8 | OLI | 30 m | 1754 |

**2.2 Generating lake inventories 1984–2019**

Image composites were classified in ArcGIS Pro using an object-based supervised classification (Support Vector Machine). Manual training samples were selected for snow, ice, water, bedrock, supraglacial debris, and vegetation for each individual time step as snow, ice, and water can vary from year to year. To reduce false positives from mountain shadows, which have a similar spectral signal as water (Fig. S2), a slope threshold of 10° was implemented (Zhang et al., 2018; Shugar et al., 2020;

Chen et al., 2021). We used a digital elevation model (DEM) composed of the national elevation dataset (NED; 10 m



resolution) and Worldview-derived DEMs (resampled to 10 m resolution; DEMs created by the Polar Geospatial Center from DigitalGlobe, Inc. imagery). A minimum area threshold of 0.05 km² (~7.5 pixels squared) excludes pixel-level noise and small lakes with minimal hydrological impact and GLOF potential (Carrivick and Quincey, 2014; How et al., 2021). Lakes with margins entirely outside a 1 km buffer from the RGI were eliminated to minimize the inclusion of lakes disconnected

from a glacial system (Shugar et al., 2020). Wet, supraglacial debris bands are often misclassified as lakes, likely due to the presence of supraglacial water and therefore similar spectral properties (Fig. S3). A length to area threshold of 0.3 was implemented to remove these long and thin features with non-typical lake shape.

All lake outlines were visually inspected and, if necessary, the lake margins were manually adjusted to produce a final delineation. Lakes were added or excluded based on visual inspection and consideration of all four time steps together; lakes

<0.05 km² were manually added back in if the lake grew in subsequent years to minimize false signals of lake formation. Every lake in each time step was then classified by i) dam type (Sect. 2.3.1), ii) topological position (Sect. 2.3.2), and iii) stability (Sect. 2.3.3).

## 2.3 Lake characteristics

### 2.3.1 Dam type classification

Using a combination of DEMs and high-resolution satellite imagery (Google Earth and ArcGIS basemap), each lake's dam type was manually classified (Buckel et al., 2018). Four different dam types were identified in this study (Fig. 2):

*i) Moraine-dammed lakes:* most frequently located at the glacier terminus, impounded behind a terminal or lateral moraine (Otto, 2019);

*ii) Ice-dammed lakes:* located along glacier margins or within tributary valleys and blocked by glacier;

*iii) Supraglacial lakes:* found on the surface of the glacier, often dammed by glacier surface topography (ice or debris) within the ablation zone; and

*iv) Bedrock-dammed lakes:* frequently located in cirques with minimal remaining glacial ice, or in other overdeepenings created from glacial erosion (Otto, 2019).





**Figure 2: Examples of typical lakes for each dam type (moraine (a-b), ice (c-d), supraglacial (d-f), and bedrock(g-h)), and changes in lake area from 1984–1988 (left; red) and 2016–2019 (right; blue). False color images using Landsat bands for shortwave infrared (SWIR), near infrared (NIR), and red.**



### 2.3.2 Topological position

In addition to dam type, all glacial lakes were classified based on their relationship to their source glacier (Nie et al., 2017;
Rounce et al., 2017), into one of the following categories:

*i) Proglacial:* lakes at the terminus of the glacier, in contact with the ice;

*ii) Supraglacial:* lakes on the surface of the glacier, most commonly within debris;

*iii) Detached:* lakes fed by glaciers but not in contact with ice;

*iv) Unconnected:* detached lakes not fed by glaciers; or

*v) Ice:* ice-dammed lakes located at ice margins or in tributary valleys.

### 2.3.3 Lake stability classification

Lake stability simply refers to whether or not a lake is present, rather than the stability of the lake shoreline. Five stability
classifications are identified within this dataset:

*i) Forms*: lakes which appear after 1984–1988 and are present in every mosaic through 2016–2019;

*ii) Forms-Drains*: lakes which form after 1984–1988 and drain by 2016–2019;

*iii) Drains*: lakes which were present in 1984–1988 and disappear by 2016–2019;

*iv) Drains-Refills*: lakes which are present in 1984–1988, are not present for one or two time steps, then reappear by 2016–
2019; or

*v) Persistent*: lakes which are present in all four time steps.

### 2.4 Lake area change

Absolute area change (ΔA) was calculated for all lakes present in the latest time step (2016–2019), taken as the difference
between the last (2016–2019) and first (either 2007–2011, 1997–2001, or 1984–1988) outline for each lake. For lakes which
first appear in the 2016–2019 composite, ΔA is equal to lake area in 2016–2019. Rate of change (km² per decade, calculated
between the midpoint of each interval) was also calculated to minimize bias of longer standing lakes having a larger area
change. Rate of change per time step was also calculated to compare rates of area change over time.

### 2.5 Error

Error in lake delineation was calculated assuming an error of ± 1 pixel for the entirety of each lake perimeter (Chen et al.,
2021):

$E = P * R$,                                                                                                          (1)





where P is the perimeter of the lake (km), R is the pixel resolution of the imagery (0.030 km for Landsat), and the resulting error (E) is in km$^2$. This error calculation is more generous than numerous lake inventories which assume $\pm$ 0.5 pixel error (e.g., Fujita et al., 2009; Salerno et al., 2012; Zhang et al., 2015; Nie et al., 2017; Rounce et al., 2017; Wang et al., 2020). Error for the difference in areas was calculated using the theory of propagation of uncorrelated error:

$$175 \quad E_{diff} = \sqrt{E_x{}^2 + E_y{}^2}. \tag{2}$$

where $E_x$ and $E_y$ are the error for the first and second lake outlines, respectively. Lakes where the area difference was greater than $E_{diff}$ were determined to have detectable area change.

## 3 Results

### 3.1 Lake evolution 1984–2019

180   The overall number of lakes in the Alaska region increased from 485 in 1984–1988 to 661 in 2016–2019, and total area grew from 817.3 km$^2$ in 1984–1988 to 1284.4 km$^2$ in 2016–2019 (Fig. 3). The distribution of lakes of each dam type has remained fairly consistent over time, with 41–44% of all lakes being moraine-dammed, 27–30% bedrock-dammed, 9–14% ice-dammed, and 16–23% supraglacial. However, the contribution of each dam type to the total lake area has changed. The contribution of ice-dammed lakes decreased (16% in 1984–1988 to 6% in 2016–2019), while moraine-dammed lakes increased (67% in 1984–

185   1988 to 79% in 2016–2019). While supraglacial lakes have increased in number over time, their contribution to total lake area has remained consistent around 2–3%. Supraglacial lakes accounted for 20% of lakes by number in 2016–2019, though they only contributed to 3% of the total area. Conversely, moraine-dammed lakes composed 39–44% of all lakes yet contributed 67–79% of the total area.



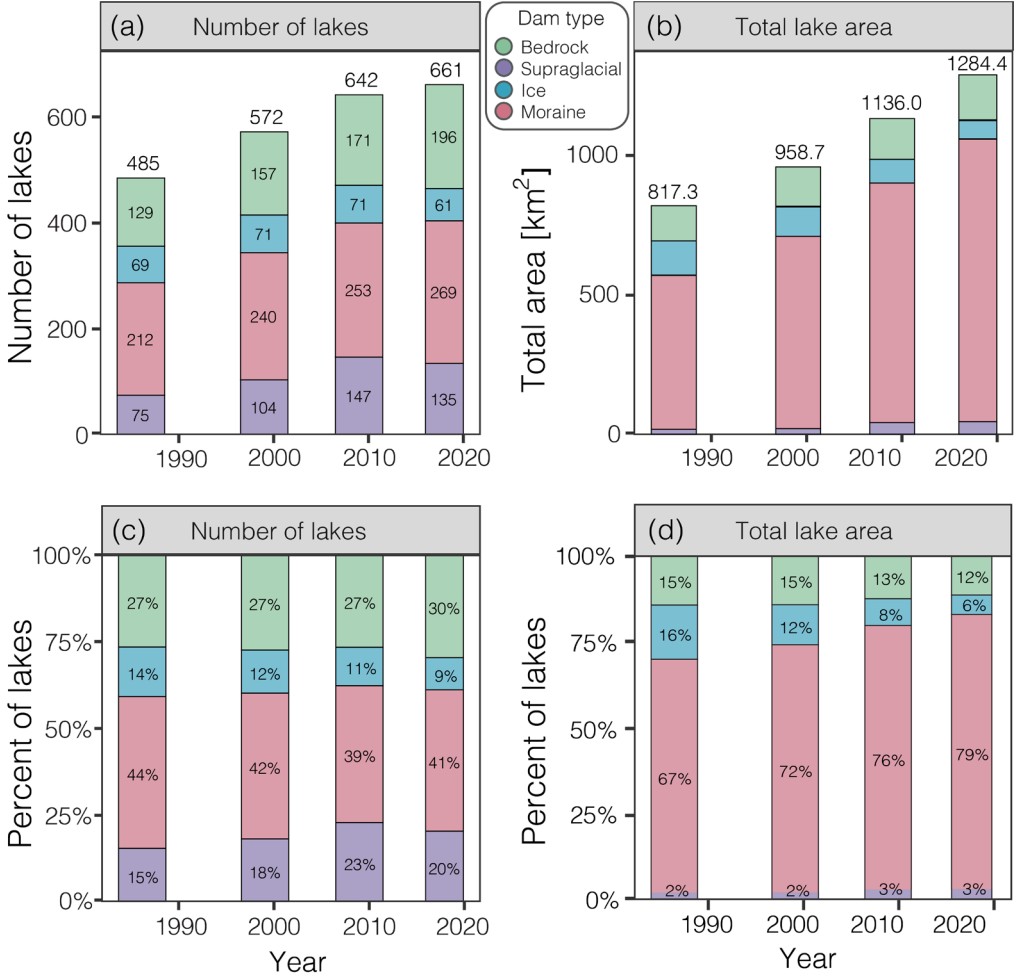

**Figure 3: Time evolution of ice-marginal lakes in terms of (a) unscaled number of lakes, (b) unscaled lake area, (c) number of lakes scaled by total number of lakes in that timestep (percent), and (d) lake area scaled by the total lake area in that timestep (percent). Lakes are colored by lake dam type (legend between panels a and b). Bar widths correspond to imagery time intervals.**

Lake stability describes whether a lake is persistent, or if it formed or drained during any of the four time intervals used for analysis. Lakes that appear in all four time intervals ("persistent lakes") dominate both the number and area of bedrock-, ice-, and moraine-dammed lakes (Fig. 4). Supraglacial lakes are the least stable, as this class is dominated by non-persistent lakes. Changes in the number of persistent lakes from year to year is due to lakes either splitting or merging. The rate of formation of new bedrock and moraine-dammed lakes appears relatively constant over time, with ~10–30 new lakes of each dam type first appearing in each time interval.





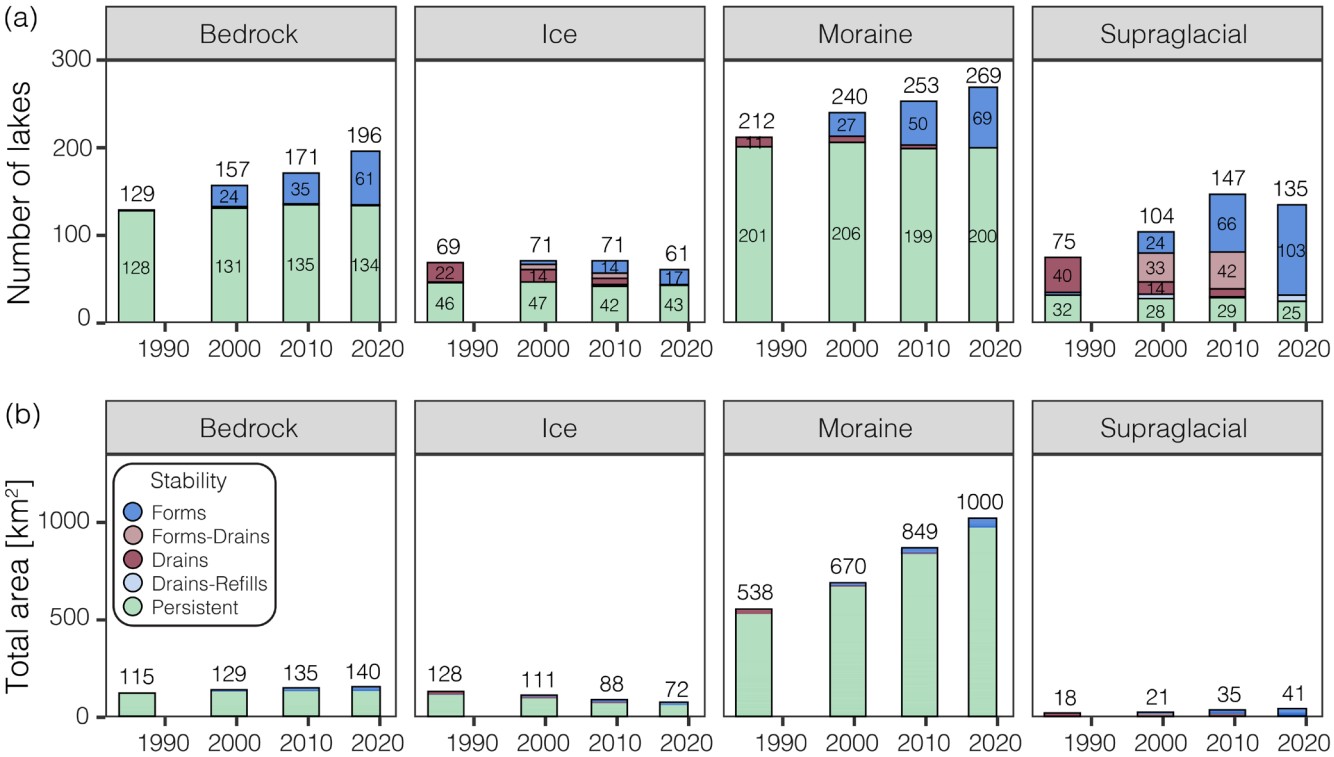

**Figure 4: Number of lakes (a) and total area (b) of each dam type through time. Bar widths correspond to imagery time intervals.**

Moraine-dammed lakes added 58 lakes (+27.5%) and grew by 468.5 km$^2$ (+85%) from 1984–1988 to 2016–2019, while bedrock-dammed lakes added 68 lakes (+52.7%) and grew by 31.8 km$^2$ (+26.3%; Fig. 4). Though they added a similar number of lakes, moraine-dammed lakes increased in total area more rapidly than bedrock-dammed lakes, primarily through the growth of pre-existing lakes.

While bedrock and moraine-dammed lakes steadily increased in lake frequency and area, supraglacial lakes exhibited more variability. This class increased by 72 lakes from 1984–1988 to 2007–2011, yet lost 13 lakes from 2007–2011 to 2016–2019, resulting in an overall increase of 59 lakes (78% increase). Despite varying frequency, supraglacial lakes grew in total area between each time period, resulting in a total increase of 22.8 km$^2$ (+127%) from 1984–1988 to 2016–2019. Supraglacial lakes had the largest percent increase in area; however, they contributed the least (2-3%) to total lake area across different dam types. Ice-dammed lakes are the only dam type with decreasing number and area, losing 9 lakes (–13%) and 56 km$^2$ (–43%) overall from 1984–1988 to 2016–2019. However, a total of 22 individual lakes drained and 16 formed throughout this time period.

Lake frequency and area change distribution varies with lake area (Fig. 5). A majority of lakes (84%) are less than 1 km$^2$, yet they compose just 13% of the total area change from 1984–1988 to 2016–2019. There are 11 lakes (10 moraine-dammed, 1 bedrock-dammed) with individual areas greater than 20 km$^2$, accounting for 250.8 km$^2$ (54%) of the total area growth from 1984–1988 to 2016–2019. Moraine-dammed lakes dominate the proportion of lakes greater than 1 km$^2$.





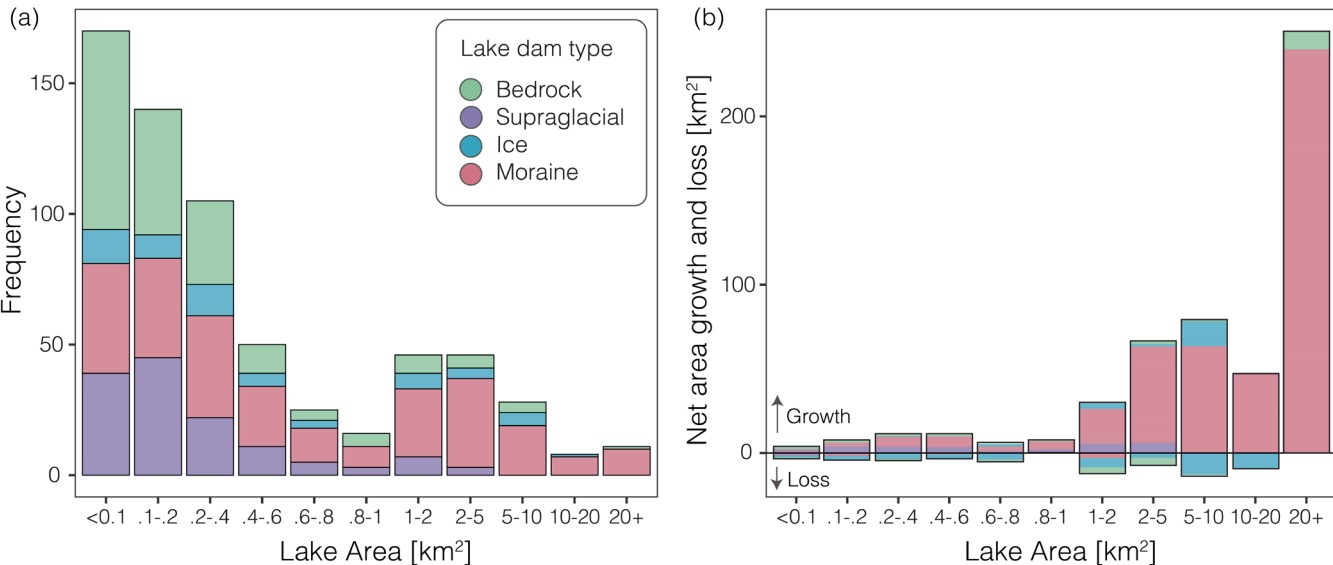

**Figure 5: Distribution of lake area frequency in 2016–2019 (a) and lake area change from 1984–1988 to 2016–2019 (b) for different sized lakes. (b) sums area loss and gain within in each dam type, showing the range of behavior within a single size class rather than total area change which masks area loss. Note: bins include varying ranges of lake area.**

### 3.2 Individual lake area evolution

From 1984–1988 to 2016–2019, 791 distinct lakes were identified, 661 of which were still present in 2016–2019. Absolute area change was calculated for each individual lake (Fig. 6). Detectable change (error < area change) occurred for 334 of the total 791 lakes (44%). Within each dam type, 63% of moraine-dammed, 45% of ice-dammed, 39% of supraglacial, and 24% of bedrock-dammed lakes experienced detectable change.



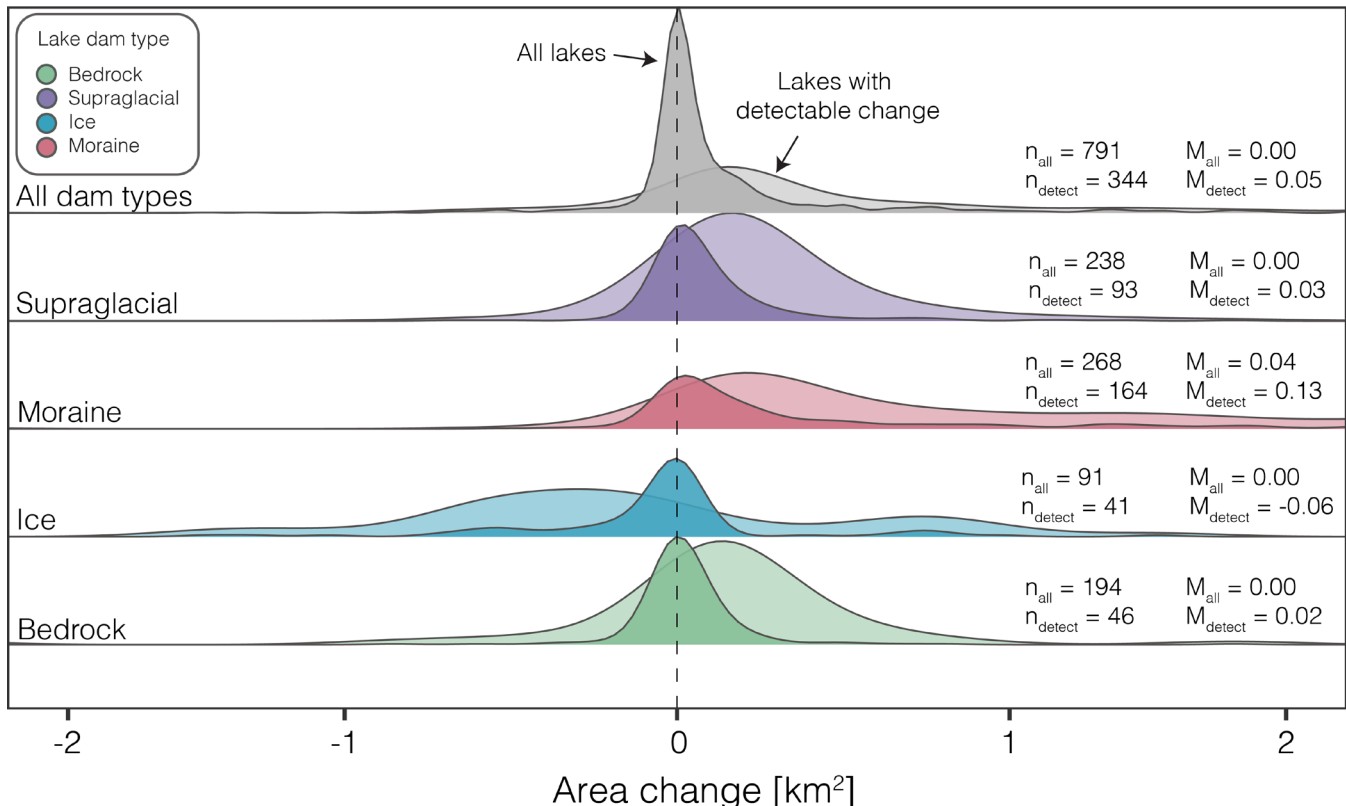

**Figure 6: Smoothed density distribution (normalized to 1 for each dam type) of absolute lake area change for all lakes (dark curves) compared to that for only lakes with detectable change (light curves) for each dam type, with number of lakes (n) and median lake area change (M). Note that the x-axis has been limited to -2 and 2 km². For the full distribution, see Fig. S4.**

The median area change for lakes of all dam types (n = 791) is 0.00 km², while the median area change for lakes of all dam types with detectable change (n = 344) is 0.05 km² between 1984–1988 and 2016–2019. When considering all lakes within each dam type, ice-dammed (n = 91), supraglacial (n = 239), and bedrock-dammed (n = 194) lakes all have a median area change of 0.00 km², while moraine-dammed lakes (n = 267) have a median area change of 0.04 km². Considering only lakes with detectable change, the median change for moraine-dammed lakes (n = 163) is 0.14 km², ice-dammed lakes (n = 41) is –0.06 km², supraglacial lakes (n = 94) is 0.04 km², and bedrock-dammed lakes (n = 46) is 0.02 km².

Proglacial lakes (moraine- and bedrock-dammed) experienced the largest median change (0.06 km² all, 0.14 km² detectable), while ice-dammed lakes are the only dam type to experience negative median change (–0.06 km² detectable; Table 2; Fig. S5). Unconnected and detached lakes have a median change of 0.00 km² when considering all lakes, and a median change of 0.02 km² for lakes with a detectable change. Proglacial moraine-dammed lakes have a wider distribution and higher median change than detached or unconnected moraine-dammed lakes (Table 2). Proglacial moraine-dammed lakes also have a wider distribution and higher median change than proglacial bedrock-dammed lakes.



**Table 2:** Count (area, km$^2$) of glacial lakes by dam type and location between 1984 and 2019, as well as median area (km$^2$) and median area change (km$^2$) for all lakes. Median area change for lakes with detectable change (km$^2$) are within the parentheses.

| Dam Type | Location | 1984–1988 | 1997–2001 | 2007–2011 | 2016–2019 | Change 1984–88 to 2016–19 | Median area | Median change |
|---|---|---|---|---|---|---|---|---|
| Moraine | Proglacial | 171 (539) | 190 (664) | 190 (839) | 197 (988) | 26 (449) | 0.71 | 0.3 (0.77) |
| | Detached | 36 (9.5) | 45 (19.2) | 58 (24.4) | 68 (28.7) | 32 (19.2) | 0.12 | 0.0 (0.08) |
| | Unconnected | 4 (2.1) | 4 (2.2) | 4 (2.0) | 4 (2.0) | 0 (-0.1) | 0.40 | 0.0 (0.0) |
| Bedrock | Proglacial | 12 (4.0) | 10 (2.7) | 13 (5.0) | 17 (7.5) | 5 (3.5) | 0.19 | 0.1 (0.19) |
| | Detached | 58 (92.7) | 76 (103) | 78 (105) | 85 (106) | 27 (13.3) | 0.13 | 0.0 (0.11) |
| | Unconnected | 59 (23.9) | 71 (35) | 80 (36.7) | 95 (38.7) | 36 (14.8) | 0.16 | 0.0 (0.09) |
| Ice | Tributary | 43 (115) | 41 (96.0) | 44 (71.4) | 36 (62.7) | -7 (-52.3) | 0.75 | 0.0 (-0.37) |
| | Margin | 26 (13.4) | 30 (14.9) | 27 (16.9) | 24 (9.57) | -2 (3.83) | 0.16 | 0.0 (-0.13) |
| Supraglacial | | 76 (17.9) | 105 (21.4) | 148 (35.1) | 135 (40.7) | 59 (22.8) | 0.12 | 0.0 (0.04) |
| **Total** | | **485 (817.3)** | **572 (958.7)** | **642 (1136.0)** | **661 (1284.4)** | **176 (467.1)** | **0.21** | **0.01 (0.21)** |

Rate of area change (km$^2$ per decade) follows a similar distribution as absolute area change (Fig. S6). When assessing all dam types together (n = 791), the median change rate is 0.00 km$^2$ per decade. The median change rate for lakes of all dam types with detectable change (n = 344) is 0.02 km$^2$ per decade. When considering all lakes within each dam type, moraine-dammed lakes are the only dam type with a change rate greater than zero (0.05 km$^2$ per decade). Supraglacial and bedrock-dammed lakes experience very small changes on a decadal scale, with a median change rate of 0 for all lakes as well as the subset of

detectable lakes. Ice-dammed lakes with detectable change experienced a median change rate of –0.02 km$^2$ per decade.

**3.3 Regional lake distribution**

Distribution of lake dam type and total lake area varies spatially by subregion (Table 3; Fig. 7), though median lake area change does not vary substantially when considering all lakes per region (ranges from 0.04–0.06 km$^2$). Ice-dammed lakes occur most frequently in the St. Elias Mountains (n = 26 in 2016–2019) and least frequently in the W. Chugach Mountains (n = 6 in 2016–

2019), Brooks Range (n = 0), and the Aleutians (n = 0). The N. Coast Ranges are dominated by moraine- and bedrock-dammed lakes, whereas the Alaska Ranges and St. Elias Mountains are dominated in number by supraglacial lakes. Supraglacial lakes occur in the regions with the highest debris cover (19% of glacier area in the Alaska Ranges, and 13% in the St. Elias Mountains; Herreid and Pellicciotti, 2020). However, supraglacial lakes contribute little to the total lake area. The number of



lakes within a region does not directly predict area; the N. Coast Ranges host the highest number of lakes (n = 234, 382.5 km$^2$),

yet the St. Elias Mountains have the greatest total lake area (n = 181, 545.5 km$^2$).

**Table 3: Number of lakes, glacierized area, debris covered area, percent debris cover, normalized lake frequency, normalized lake area, normalized lake area change, and specific glacier mass balance (from Jakob et al., 2021) per subregion.**

| Region | # of lakes 2016–19 | Total glacier area (km$^2$) | Total debris cover (km$^2$) | Total lake area (km$^2$) | Percent debris cover | Normalized lake freq (# per 100 km$^2$ ice) | Normalized lake area (km$^2$ per 100 km$^2$ ice) | Normalized lake area change (km$^2$ per 100 km$^2$ ice) | Specific mass balance (m w.e. yr$^{-1}$) |
|---|---|---|---|---|---|---|---|---|---|
| Brooks Range | 11 | 346 | 10.3 | 0.7 | 3.0 | 3.2 | 0.2 | 0.1 | N/A |
| Alaska Ranges | 103 | 16278 | 3135.5 | 143.0 | 19.3 | .63 | 0.9 | 0.2 | -0.41 ± 0.05 |
| Aleutians | 25 | 1912 | 198.7 | 30.0 | 10.4 | 1.3 | 1.6 | 0.6 | -0.64 ± 0.10 |
| Chugach Mtns | 107 | 12052 | 586.2 | 182.8 | 4.9 | 0.9 | 1.5 | 0.6 | -0.80 ± 0.09 |
| St. Elias Mtns | 181 | 33174 | 4446.3 | 545.5 | 13.4 | 0.5 | 1.6 | 0.6 | -1.03 ± 0.10 |
| N Coast Ranges | 234 | 22963 | 1553.8 | 382.5 | 6.8 | 1.0 | 1.7 | 0.6 | -1.08 ± 0.09 |






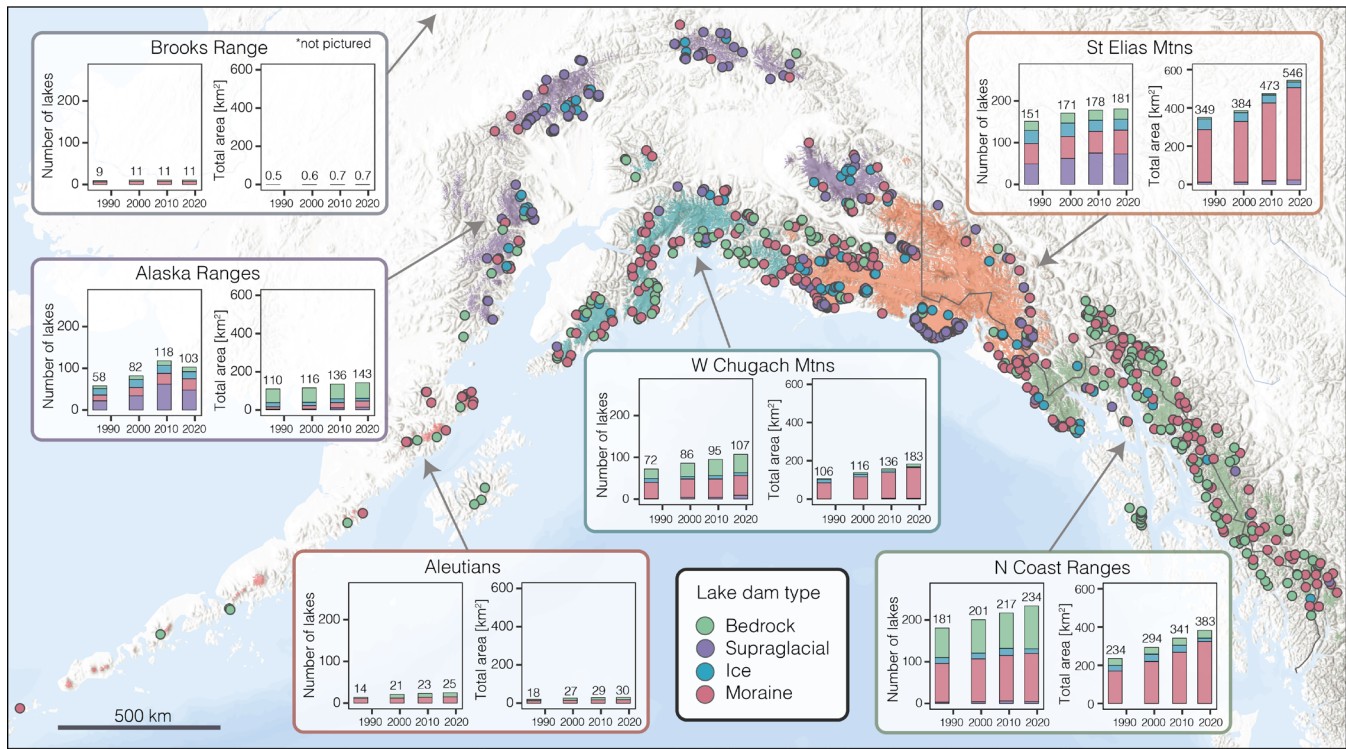

**Figure 7: Dam type distribution (spatial, frequency, and total area) for each Alaska subregion. Note: Brooks Range not pictured on map- see Fig. 1. Basemap provided by ESRI, 2021.**


Persistent lakes dominate the total area for each subregion, likely because new lakes tend to be small. The N. Coast Ranges have the most persistent lakes (n = 175 in 2016–2019), with 57 lakes forming yet contributing little to the total area (Fig. S7). Changes in persistent lake numbers between time steps is due to lakes splitting or coalescing through time. The Alaska Ranges and St. Elias Mountains have the largest number of lakes that ultimately drain, likely due to the high frequency of supraglacial

lakes in these regions and their spatial and temporal heterogeneity.

## 4 Discussion

Our inventory of ice-marginal lakes in Alaska from 1984 to 2019 demonstrates that lake area changes vary based on lake dam type and a lake's topological position. By tracking individual lakes through time and categorizing lakes by dam type, region, and spatial relationship to their source glacier, we are able to parse trends that would otherwise be overlooked if lakes within

Alaska were considered as a whole.

Overall, ice-marginal lakes in Alaska have grown in both number and area between 1984 and 2019, following the trend in both regional (Wang et al., 2013; Carrivick and Quincey, 2014; Nie et al., 2017; Song et al., 2017; Emmer et al., 2020) and global (Shugar et al., 2020) studies. However, ice-dammed lakes have decreased in number and area whereas moraine-dammed lakes





have increased at a faster rate than the average when considering all dam types together. By examining individual lakes, we
identify that the majority of lakes have not experienced a detectable area change over the period of study, and that 11 large,
primarily moraine-dammed lakes contributed to 54% of total area growth. Analyzing all dam types together would fail to
identify decreases in number and area of ice-dammed lakes, underestimate the increases in proglacial moraine-dammed lakes,
and likely overestimate the importance of the appearance of small, supraglacial lakes.

## 4.1 Alaska's ice-marginal lakes

Lake location and dam type, which are often linked, provide a simple, physically-based metric for parsing ice-marginal lakes
in Alaska. The majority of moraine-dammed lakes are proglacial, located behind a Little Ice Age (LIA; ~1250–1850 AD)
moraine at the front of a retreating glacier (e.g., Wiles et al., 1999; Santos and Córdova, 2009; Solomina et al., 2015, 2016).
They tend to be associated with clean-ice glaciers (82%), defined by Brun et al. (2019) as <19% debris cover (Fig. 8). Moraine-
dammed lakes occur with both large and small glaciers, as most temperate glaciers can form a moraine during a period of
advancement (e.g., the LIA), as long as sediment supply to the terminus is high. Basin geometry and glacier hypsometry likely
play a large role in determining the lake expansion rate and maximum lake area, as the underlying bed slope influences how
quickly the glacier retreats (depending on glacier mass balance; see Sect. 4.5). Basin geometry also determines the maximum
lake level and the point at which the glacier detaches from the lake.





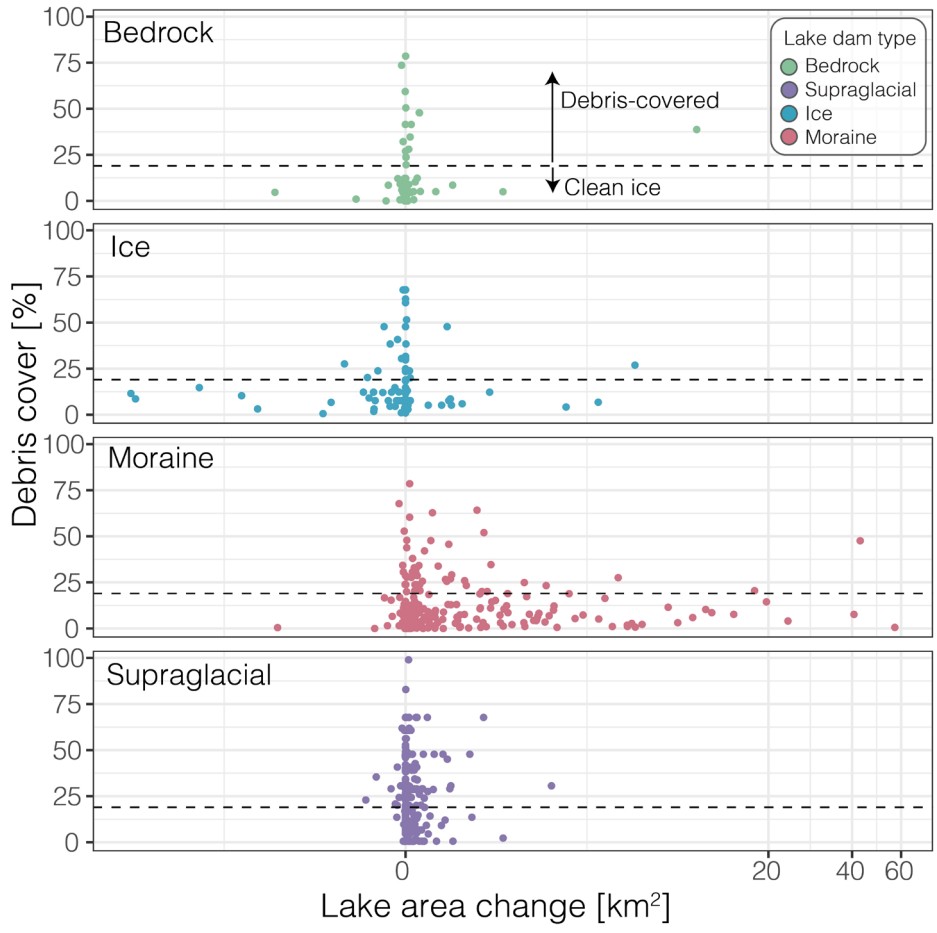

**Figure 8. Percent glacier debris cover vs lake area change (km²) for lakes associated with glaciers > 2 km². Each dot represents an individual lake. Dashed line is at 19%, above which is considered debris-covered ice and below is considered clean ice (Brun et al., 2019).**

Ice-dammed lakes are primarily located next to clean-ice glaciers (65%), with larger, lower slope areas and positive hypsometric indices (bottom-heavy; Fig. 9). The largest lakes occur in tributary valleys, dammed by main branch ice, while many small lakes are found in pockets between glacier margins and valley walls. The decrease in ice-dammed lake number and area is likely due to the down-wasting of glacier surfaces (e.g., Larsen et al., 2015; Jakob et al., 2021), decreasing the height of the ice dam and therefore decreasing maximum area (and volume) of the lake (e.g., Kienholz et al., 2020). The formation of new conduits alongside or beneath the glacier could also be influencing ice-dammed lake drainage (Post and Mayo, 1971). Though not explicitly examined in our study, ice thinning was the primary factor in the loss of an ice dam for 82% of lakes lost from land-terminating glaciers and 62% of loss from lake terminating glaciers between 1971 and 2008 (Wolfe et al., 2014).





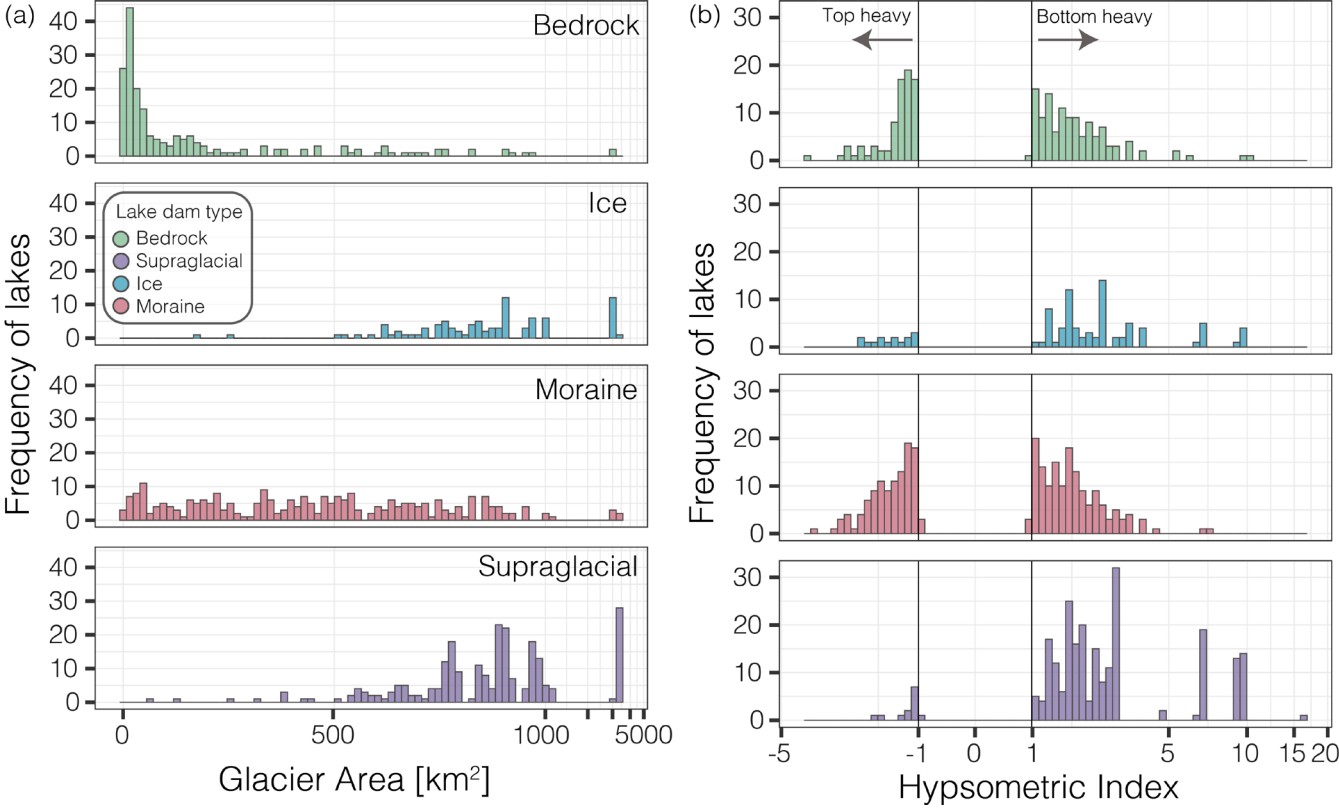

**Figure 9. Histogram of number of lakes and associated glacier area, separated by dam type (a) and histogram showing number of lakes within each hypsometric index (HI) bin, separated by dam type (b). Hypsometric index was calculated by McGrath et al. (2017). HI values greater than 1 indicate a bottom heavy glacier, HI values less than -1 indicate a top heavy glacier. Note the logarithmic x-axis scales.**

An overwhelming majority (>80%) of bedrock-dammed lakes are associated with clean-ice glaciers less than 10 km² in area (Figs. 8 & 9). These types of lakes generally form within cirques, where a small glacier has retreated and exposed an overdeepening. Bedrock-dammed lakes can indicate an advanced stage of lake development (Emmer et al., 2020), such as is found in parts of the N. Coast Ranges. Most bedrock-dammed lakes (>90%) are either detached or unconnected, with minimal influence from ice-water contact. The small changes in area of unconnected bedrock-dammed lakes could be due to changes in the regional water balance rather than glacier dynamics.

Supraglacial lakes generally form on low-sloped debris-covered tongues of glaciers (Reynolds, 2000), where thin debris enhances melt and thick debris insulates ice and reduces surface melt (Östrem, 1959), often leading to heterogeneous surficial topography. Debris cover tends to reduce ice discharge and surface melt, leading to longer glaciers (Anderson and Anderson, 2016). The inventoried supraglacial lakes in Alaska follow these trends, with their distribution strongly skewed towards flat, bottom-heavy glaciers, as is characteristic of debris-covered glaciers (Reynolds, 2000). These lakes can occur on glaciers with any amount of debris cover, though have the highest concentration (64%) out of any lake dam type on glaciers with >19%



debris cover (Fig. 8). A majority of supraglacial lakes are located on a few large, heavily debris covered glaciers. For example, 95% of supraglacial lakes in the St. Elias Mountain region are located on two large, piedmont glaciers (Malaspina and Bering glaciers). Although these lakes are abundant, they exhibit high spatiotemporal variability and contribute minimally to total

lake area (2–3%; Fig. 4).

Although additional factors are required to comprehensively assess GLOF hazards, dam type and lake volume are primary indicators of likelihood of a dam to fail and the potential magnitude of a flood. Lake volume scales with area (e.g., Huggel et al., 2002; Cook and Quincey, 2015), and peak flood discharge scales with volume (e.g., Walder and Costa, 1996; Veh et al., 2020). Ice-dammed lakes are known to undergo multiple fill/drain cycles with the capability of producing multiple GLOFs

(e.g., Anderson et al., 2003; Jacquet et al., 2017), whereas moraine-dammed lakes tend to only GLOF once, as their dam is typically compromised in the process. The observed decrease in number and area of ice-dammed lakes in Alaska suggests an overall decrease in GLOF hazards from these types of lakes, although the specifics of remaining lakes need to be evaluated on a case by case basis (e.g., Kienholz et al., 2020). The increase in number and area of moraine-dammed lakes provides a greater number of potential source lakes and a larger potential flood volume, however, factors such as surrounding slope stability (e.g.,

landslides, rockfalls, permafrost), moraine geometry and stability (e.g., slope, presence of an ice core), and downstream impacts must be evaluated first to determine GLOF potential and hazard (e.g., Worni et al., 2013; Rounce et al., 2017). As the nature of the dam type influences lake stability, the opposing trends of ice-dammed and moraine-dammed lakes complicates a region-wide assessment of changing GLOF hazards in the Alaska region.

## 4.2 Regional trends

The Alaska region of the RGI is separated into six second-order regions (see Fig. 8), used here as subregions for analysis. Lakes are most numerous in the N. Coast Ranges (35% of all lakes) and St. Elias Mountains (27%), though these ranges also have the largest glacier area (24% and 41% of total ice in Alaska, respectively). Normalized lake frequency (number of lakes per 100 km$^2$ of glaciers) is highest in the Brooks Range, though it has the lowest normalized lake area (lake area per 100 km$^2$ of glaciers; 0.2) as total lake area sums to just 0.7 km$^2$ (Table 3). We interpret these normalized lake statistics as reflecting the

late stage of ice-marginal lake development, minimal LIA glacier advance, as well as the relatively small glacier extent in the Brooks Range even during the last glacial maximum (Kaufman and Manley, 2004). Normalized lake area is highest in the N. Coast Ranges (1.7 km$^2$ per 100 km$^2$ of glaciers) due to the abundance of smaller glaciers and many proglacial lakes. The St. Elias Mountains have the largest total lake area; however, it also contains large icefields, which results in a similar normalized lake area as the Aleutians and the Chugach Mountains (1.5–1.6 km$^2$ per 100 km$^2$ of glaciers). This suggests that the area of

ice-marginal lakes scales with glacier area in the coastal ranges (i.e., excluding the Brooks Range and Alaska Ranges), with normalized lake area ranging from 1.5 to 1.7.

Accounting for lake area change, the Brooks Range and Alaska Ranges have a small normalized lake area change (0.1–0.2; lake area change between 1984 and 2019 per 100 km$^2$ of glaciers). The Chugach, Aleutians, St. Elias, and N. Coast Ranges, however, each have a higher normalized lake area change (0.6). These numbers suggest a dichotomy in which coastal glacier-





lake systems are changing faster than those in more interior settings, in agreement with recent work investigating physical controls on ice-marginal lake area change (Field et al., 2021). The Alaska Ranges have the highest percent debris cover (19%), which could be a factor in the low normalized lake area and normalized lake area change seen in this region. Supraglacial lakes are the most abundant dam type in the Alaska Ranges (47% in 2016–2019), characterized by small areas and area changes (Table 2).

**4.3 Temporal trends**

Rate of lake area change (km$^2$ per decade) through time indicates whether ice-marginal lake evolution has remained constant between different time periods, or whether the lakes have experienced periods of accelerated or decelerated change. Bedrock dammed lakes experienced little variation in rate of area change throughout the period of study (Fig. 10), with a rate that hovers around zero, likely due to the fact that a majority (>90%) of bedrock-dammed lakes are either detached from their associated

glacier or unconnected to any glacier. Ice-dammed lakes consistently experienced a negative median rate of area change, with the most negative rate of growth between 2007–2011 and 2016–2019. This implies drainage of ice-dammed lakes may be accelerating, which could be due to an increased rate of glacier thinning since 2013–2014 (Jakob et al., 2021), though this link requires further investigation. Supraglacial lakes exhibit the opposite trend, growing at a faster rate after 2007–2011 than they did prior. Moraine-dammed lakes experienced the largest median rate of change between 1984–1988 and 1997–2001. A

decreasing median rate of change for moraine-dammed lakes yet large total area growth is likely due to the growth of a few large, outlier lakes whose influence is minimized by comparing median rather than mean rates of change. Proglacial lakes experienced the overall largest rate of growth, even when compared to all moraine-dammed lakes, suggesting that topological position, rather than strictly dam type, is a better indicator for rate of ice-marginal lake area change. Together, these data suggest the ice-marginal lakes of Alaska and northwestern Canada could be undergoing a transition, with the dominant

mechanism for regional lake area change shifting from moraine-dammed lake expansion to growth and coalescence of supraglacial lakes in the future.





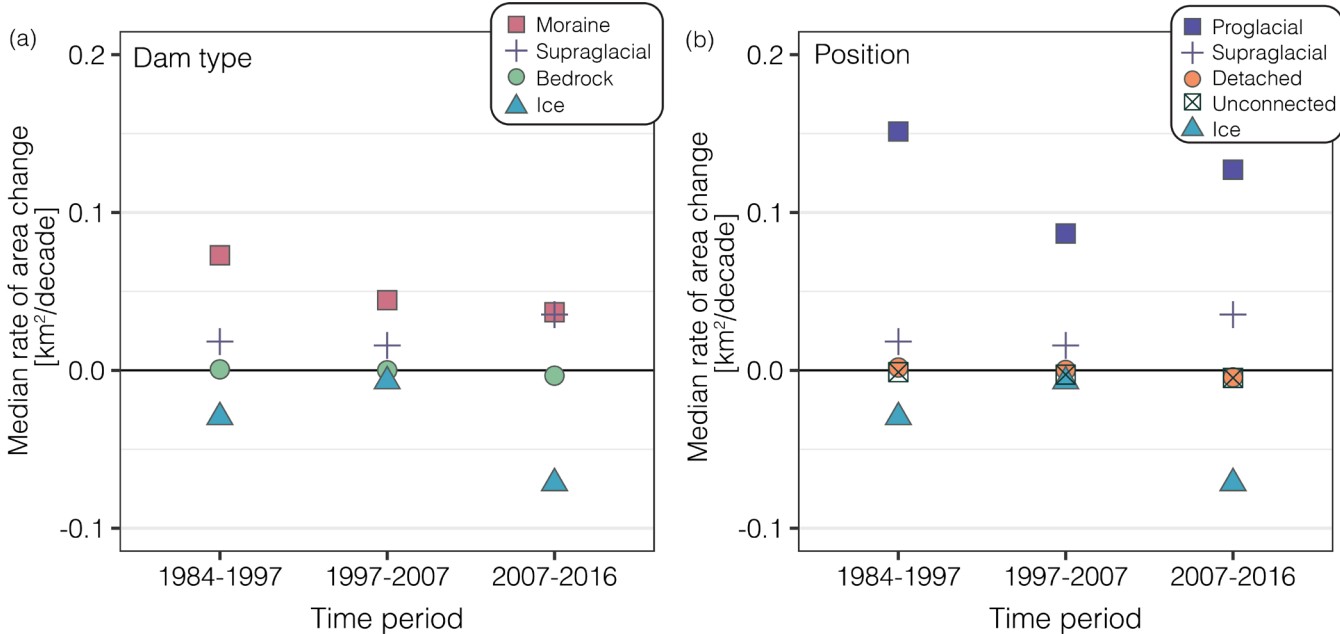

**Figure 10. Median rate of area change between each time step, categorized by dam type (a) and topological position (b).**

**4.4 Comparison to other regions**

Alaska's glaciers cover a large spatial extent, spanning different climatic, geologic, and glacial environments. This makes the region particularly unique in relating to various regions around the world. The heavily debris-covered areas (e.g., Alaska Ranges) have similarities to the Himalaya, areas of advanced clean-ice glacier retreat (e.g., N Coast Ranges) are more similar to the Peruvian Andes, and the large and complex glaciers/icefields (e.g., St. Elias Mountains) could relate to areas such as the Patagonian Icefields. Trends within these Alaska subregions could infer characteristics of ice-marginal lakes in comparable

mountain ranges, and vice versa.

Many studies on ice-marginal lakes concentrate on High Mountain Asia (e.g., Nie et al., 2017; Zhang et al., 2018; Veh et al., 2020; Wang et al., 2020; Chen et al., 2021), a region with abundant proglacial and supraglacial lakes. Alaska and the Himalaya regions (RGI regions 14, 15) have a similar percent debris cover (~15%), though Alaska contains larger glacier and debris-covered areas (Herreid and Pellicciotti, 2020). Herreid and Pellicciotti (2020) calculated a debris stage for each RGI region

from 0 to 1, where 1 indicates the ablation zone is entirely covered in debris. Debris stage is slightly higher in the Himalaya (~0.6–0.7) than Alaska (~0.5), though Alaska has a higher debris expansion potential (Herreid and Pellicciotti, 2020). This suggests that as debris cover expands in Alaska, these areas could become more similar to the heavily debris-covered areas in the Himalaya. Supraglacial lakes in the Himalaya vary considerably in location, shape and size, appearing, disappearing, and coalescing through time (Benn et al., 2001; Gardelle et al., 2011; Nie et al., 2017), behavior exhibited in Alaska as well. These

similarities indicate that supraglacial lakes within these two regions have similar characteristics, and that understanding one region could aid in understanding the other.



Glacierized mountain ranges around the world exhibit varying stages of glacial retreat. The Peruvian Andes are considered to be in a late stage of glacier retreat with little potential for further lake expansion, as ice generally remains in steep, high elevation basins (Emmer et al., 2020). Though the Cordillera Blanca contains tropical glaciers, their topographic locations and areas mimic glaciers within the N. Coast Ranges. In Peru, glaciers have retreated far off their LIA moraines, such that the majority of proglacial lakes shifted from moraine dammed in 1948 to bedrock dammed in 2017 as glaciers retreated and exposed bedrock overdeepenings (Colonia et al., 2017; Emmer et al., 2020). Though ice remains in many main valleys in the N. Coast Ranges, glaciers are smaller, steeper, cleaner, and less complex than other subregions in Alaska, apart from the Juneau Icefield. The abundance of moraine dammed proglacial lakes and increasing proportion of bedrock dammed lakes (39% in 1984–1988 to 44% in 2016–2019) suggests the N. Coast Ranges could be following a similar trajectory as the Cordillera Blanca, and therefore a continuing shift in dominant dam type from moraine to bedrock could be expected into the future.

Ice-marginal lake increases have been documented throughout the world, though characteristics such as dam type, lake location, and associated glacier behavior vary widely. Examining ice-marginal lake behavior at different stages in various mountain ranges can help predict how these lakes might behave in the future.

## 4.5 Feedbacks on glacier mass balance

Lake geometry likely dictates the impact a proglacial lake has on glacier mass balance. Large, deep lakes can impede a glacier from reaching equilibrium by holding the terminus at lake elevation until it has retreated out of the lake basin (Larsen et al., 2015). Ice-water contact can also enhance calving, resulting in mass loss and lake expansion through terminus retreat (Carrivick and Tweed, 2013). Larsen et al. (2015) found that lake-terminating glaciers in Alaska have a more negative median mass balance in coastal regions where large, proglacial lakes have developed. Lake-terminating glaciers also experienced more rapid thinning near their terminus than land-terminating glaciers, and appear to be less directly coupled to climate variations (Larsen et al., 2015). In the Himalaya, lake-terminating glaciers have been linked with increased glacier mass loss through enhanced terminal retreat and surface lowering (King et al., 2019; Maurer et al., 2019), and center flow line velocities that are more than twice that of glaciers which terminate on land (Pronk et al., 2021). We therefore expect that as glaciers continue to retreat in Alaska, proglacial lake presence, formation, or detachment will influence glacier mass balance, increasing thinning and glacier velocity where lakes are present.

Regional glacier thinning and mass loss rates appear to mimic lake area change in parts of Alaska. The N. Coast Ranges and St. Elias Mountains are thinning ($-1.27 \pm 0.11$ m yr$^{-1}$ and $-1.21 \pm 0.12$ m yr$^{-1}$, respectively) and losing mass ($-1.08 \pm 0.09$ m w.e. yr$^{-1}$ and $-1.03 \pm 0.10$ m w.e. yr$^{-1}$) at a greater rate than other subregions from 2011–2019 (Jakob et al., 2021). These regions also experienced the largest normalized lake area growth (see Sect. 4.2). The high thinning rate in the Aleutians ($-0.75 \pm 0.06$ m yr$^{-1}$) does not appear to be reflected in the low normalized lake area growth documented in our survey. This is likely due to the fact that 60% of lakes in the Aleutians subregion are either detached or unconnected and therefore not directly influenced by glacier dynamics. The Alaska Ranges are thinning ($-0.48 \pm 0.06$ m yr$^{-1}$) and losing mass ($-0.41 \pm 0.05$ m w.e.



yr[-1]) at the lowest average rates per region (Jakob et al., 2021) likely due to extensive debris cover, and is accompanied by the
lowest normalized lake area change outside the Brooks Range.

**4.6 Future change in ice-marginal lakes**

As global temperatures continue to rise and glaciers thin and retreat (Zemp et al., 2019), Alaska's glaciated landscape will change, as will the nature of ice-marginal lakes. Alaskan glaciers have retreated off their LIA moraines, and therefore it is unlikely that new moraine-dammed lakes will form without a sustained period of glacial advance or surge. However, proglacial
lake expansion is expected to continue as glacier retreat accommodates lake growth, dependent on basin geometry. Using the Cordillera Blanca as an analog for later stages of lake development (Emmer et al., 2020), we hypothesize a future shift in new proglacial lakes to mostly bedrock dammed lakes, as glaciers retreat into higher, steeper terrain (e.g., Linsbauer et al., 2016; Furian et al., 2021), though it is uncertain how long the present stage of moraine-dammed lake growth will persist before glaciers leave their terminal overdeepenings. Alaska's glacier complexity and relative abundance of ice-dammed lakes poses
an interesting question as to whether these lakes will continue to drain as they have since 1970 (Wolfe et al., 2014), or whether new ice-dammed lakes will form if tributary valley ice retreats faster than main branch ice. Modeling basin geometry and future ice extent would provide insight into where ice-marginal lakes in Alaska are likely to form in the future.

**5 Conclusions**

As the largest glacierized region outside the ice sheets, characterizing Alaska's ice-marginal lakes and their evolution has
important implications for glacier mass balance, ecosystem dynamics, and GLOF hazards. Varying topographic settings (valleys, cirques, debris, icefields) and a wide range of glacier complexities allow for an abundance of ice-marginal lakes of varying dam type and location. Overall, ice-marginal lakes in Alaska have grown in both number (+176, 36% increase) and area (+467 km$^2$, 57% increase) between 1984–1988 and 2016–2019, however, we demonstrate that lakes with different dam types and topological positions behave differently. Moraine-dammed, proglacial lakes experienced the largest growth due to
glacier thinning and retreat, while ice-dammed lakes experienced an overall loss in both number and area, likely due to thinning or disappearance of ice dams. Our study demonstrates the need for the inclusion of dam type and topological position in comprehensive regional ice-marginal lake studies, cautioning that analyzing all lakes together aliases patterns exhibited by each dam type, location, or subregion. These characteristics are crucial when assessing GLOF hazard potential in mountainous regions, and time-varying inventories of ice-marginal lakes are a critical first step in predicting future lake evolution.

*Data Availability.* The lake inventory is available for download at arcticdata.io/catalog/view/doi:10.18739/A2MK6591G. Additional code and data available on request.

*Author Contributions.* DM and BR designed the study. BR performed the inventory, analyzed the data, and prepared the manuscript. DM, WA, and SM contributed to data interpretation and provided manuscript comments and revisions.



*Competing Interests.* The authors declare that they have no conflict of interest.

*Acknowledgements.* This material is based upon work supported by the National Science Foundation Graduate Research
Fellowship awarded to BR under Grant No. 006784-00002. DEMs provided by the Polar Geospatial Center under NSF-
OPP awards 1043681, 1559691, and 1542736. We appreciate the productive discussions and method considerations from
Anton Hengst. We also thank Freddy Tremblay for digitizing the Post and Mayo (1971) ice-dammed lake outlines.

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
