# Peer review of "Dam type and lake position characterize ice-marginal lake area change in Alaska and NW Canada between 1984 and 2019"

_The Cryosphere, 2021_

## Referee Comment (RC2)

**Comments on "Dam type and topological position govern ice-marginal lake area change in Alaska and NW Canada between 1984 and 2019"**

The authors have completed the first comprehensive inventory of Alaskan glacier marginal lakes, sub-dividig them both by dam type, topological relationship to the glacier, and size. This is a useful approach and as a baseline the number of lakes in each category and their respective area changes is useful.  At present the paper does not provide the reader with sufficient context to understand the unique nature of many of the ice marginal lakes in Alaska, particularly the largest.  There is a lack of referencing of previous studies that have explored specific areas identifying the relationship of the lakes and glaciers in away that would lend much better context to the inventory. An over reliance on references to the Himalayan and Peruvian Andes, which are not the best or in most cases even useful analogs underscores this issue. For Alaska it is indicated that large glacier lakes have an area greater than 10 km$^2$, whereas inventories of both Cordillera Blanca and High Mountain Asia often use 0.1 km$^2$, a two order of magnitude difference in scale (Emmer et al 2016; Chen et al 2021). The combination of these issues limits the value of the inventory data.

There are several common features of the Alaskan lakes that are unusual leading to different behavior than for most glaciated alpine regions. I will review another of examples that illustrate this with referencing where appropriate.  These illustrations are meant as examples, and not specific ones the authors may choose to use or need to specifically address.

An examination of Figure 5 illustrates how context is vital. The frequency of lakes is broken down by actual area in 2016-2019 in panel A, with the largest four groups greater than 2 km$^2$ in area representing few of the total, but dominating the spatial area change noted in panel B.  There are a significant number of Alaskan glacier lakes with an area greater than 20 km$^2$. There are some specific unique characteristics of these larger lakes.

Most of the largest Alaskan glacier lakes are found in a coastal plain environment and are impounded by a coastal shoreline systems and/or proglacial deltas formed when the glacier terminated at the lake margin and outwash plains more than by moraine.  They can be categorized as moraine dammed. But there is no potential for a dam failure to cause issues at near coastal lakes such at Mendenhall Glacier, Yakutat Glacier, Excelsior Glacier or Bear Glacier. All of these lakes are less than 20 m in elevation. Johnstone Lake the proglacial lake of Excelsior Glacier it is less than 1 km from the ocean with the highest elevation between tidewater and the lake being ~10 m (Figure 1).  This lake has expanded from 9-18 km$^2$ from 1994-2018, but still poses no GLOF risk. At Bear Glacier terminates in expanding Bear Glacier Lake, which is 0.5 km from the ocean with a maximum elevation of ~7 m between the ocean and the lake. Malaspina Lake (~85 km$^2$) is the proglacial lake at the southeast margin of Malaspina Glacier.  It is 2 km from the ocean with a maximum elevation of about ~10 m. Grand Plateau Glacier Lake (~45 km$^2$ Figure 2) is in a similar position.

Many of these lakes have filled basins developed by the loss of the piedmont lobe terminus formed when the glacier exited the mountains onto low slope forelands/coastal zones. Giffen et al (2014)

examined retreat and expanding lakes of glaciers in the Kenai and Katmai regions. ie. Bear, Fourpeaked, Spotted (Figure 3), Hallo Glaciers are examples of glacier lakes filling former piedmont lobe depressions. Mendenhall, Grand Plateau are examples from southeast Alaska.  This is not a proglacial lake type seen in High Mountain Asia or the Andes north of Patagonia. Should you separate out this group of moraine dammed proglacial lakes terminating in a coastal environment? Loriaux and Casassa (2013) found a  total lake area of the Northern Patagonia Icefield of 167.5± 8.4 km$^2$ for 2011, an increase of 64.9% from 1945  (101.6±19.1 km2). They noted an 18 km$^2$ expansion of San Quintin Lake accounting for 27% of the total expansion.  This also is a coastal environment piedmont lobe terminus depression filled proglacial lake.

The size of the lakes combined when depth also allows for the production of large tabular icebergs, such as seen at Excelsior, Ellsworth, Field and Yakutat in recent years. Trussel et al (2013) examined the rapid retreat of Yakutat Glacier in Harlequin Lake (in 2020 ~70 km$^2$), with sections going afloat (Figure 4 and 5).  They note this ability the ability to calve large tabular icebergs is largely due to the limited subaqueous melt compared to tidewater glaciers. The buoyancy also requires substantial water depth. The other unique element to these lakes given the large glaciers that feed them and the coastal setting, is they can be much deeper than examples from the Himalaya or Peruvian Andes.  Trussel et al (2013) reported a depth of 325 m at the 2010 calving front of Yakutat Glacier. Loso et al (2021) note water depths of 330 m in Alsek Lake. For ths inventory when do you draw the line between a glacial lake and one that is sufficiently tidally fed to not be a lake?  Is Vitus Lake in front of Bering Glacier a part of the inventory?

For Ice Dammed lakes it has to be emphasized that there are many that have had a long history of consistent GLOF events. Neal (2007) provides a detailed examination of glacier lake outbursts from beneath Tulsequah Glacier into the Taku River.  They report on 41 GLOF's during the 1987-2004 period with the main source migrates from Tulsequah Lake to Lake No Lake, with little change in maximum outburst discharge. After 1984 a large proglacial lake has formed at the terminus of the glacier as well, which can help mitigate peak flows (Figure 6) (Pelto, 2017). Carrivick and Tweed (2016) list Tulsequah as the glacier lake with the most outburst floods. Wilcox et al (2014) report on the migration of Ice Lake which is ~17 km upglacier.  This lake drains once or twice each year usually in late summer or early fall.  This is another attribute worth noting that if a ice dammed lakes or supraglacial lakes drain into large proglacial lakes, there impact on discharge will be limited. Lake Linda on Lemon Creek Glacier is at the head of the glacier, and typically drains near mid-summer and does not refill.

Pelto et al (2013) examined the expansion of nine lakes at distributary termini of Brady Glacier that are a combination of ice-dammed and proglacial and expanded from 8.5 km$^2$ in 1948 to 18.2 km$^2$ in 2010 (Figure 7 and 8). How did you classify these lakes? Capps et al (2011) identified a depth of 200 m for Abyss Lake.  Brady Glacier an exceptional case having so many proglacial calving terminus fronts.

Merging of lakes can also occur for these large lakes.  Loso et al (2021) note that glacier thinning in the region can alter which lake a glacier feeds in this case Alsek Lake (~60 km$^2$) and Grand Plateau Lake (Figure 3). Another future merging of lakes will be at Fingers Glacier. The merging of lakes has been observed at Melbern Glacier (Clague and Evans, 1993) Gilkey Glacier (Figure 9) and Llewellyn Glacier (Pelto, 2017).

[Figure]

**Figure 1.** Excelsior Glacier retreat from 1994 to 2018 in Landsat images from 1994, 2011 and 2018. The red arrow is the 1994 terminus location and the yellow arrow is the 2018 terminus location. Point A and B are on the south and northwest side of the eastern tributary glacier (Roan Glacier). Johnstone Bay is at bottom of images.

[Figure]

**Figure 2.** Grand Plateau Glacier in 1984 and 2015 Landsat images. Red arrow is the 1984 terminus location and yellow arrows the 2015 terminus locations. D=Distributary tongue, NF=North Fork, N=Nunatak, GP=Grand Plateau. Alsek Lake top left.

[Figure]

Figure 3. Spotted Glacier in USGS map before lake formation and in 2013 Landsat image. Red arrow indicates the 2013 east side of the terminus, the pink arrow a rock knob adjacent to the 1985 terminus, and the yellow arrow a peninsula that should become an island as the further retreat occurs.

[Figure]

Figure 4. Yakutat Glacier, Alaska in 1999 and 2020 Landsat image illustrating expansion of Harlequin Lake by 40.5 km². Yellow line is the 1999 margin, orange line is the 2020 margin, and yellow dots indicate the margin of the lake shoreline. Point A indicates the 1987 terminus location, Point X and Y the 1999 terminus location. Main terminus now extends south near Point C. Northern terminus extends west from Point B.

[Figure]

**Figure 5. Landsat images from 2013 with terminus indicated by yellow dots. Point A indicates the 1987 terminus location. Note large area of melange and icebergs from the rapid disintegration during the 2010-2013 period.**

[Figure]

**Figure 6. Tulsequah Glacier in 1984 and 2017 Landsat images. The 1984 terminus location is noted with red arrows for the main terminus, northern distributary tongue which also dams Lake No Lake, southern distributary red arrow indicates ice dammed Tulsequah Lake margin. The yellow arrows**

indicate the 2017 glacier terminus locations. The retreat of 2900 m since 1984 led to a lake of the same size forming. Purple dots indicate the snowline.

[Figure]

**Figure 7. Comparison of Brady Glacier in 1986 and 2016 Landsat images. The snowline is similar in May 2016 and August 1986. Lakes noted are: A=Abyss, B=Bearhole, D=Dixon, N=North Deception, O=Oscar, Sd=South Dixon, Sp=Spur, T=Trick.**

[Figure]

Figure 8. Oscar Lake growth on the east margin of Brady Glacier in Landsat images from 2000-2020. Point A indicates glacier tongue that becomes iceberg. Blue arrows indicate flow direction.

[Figure]

Figure 9. Gilkey Glacier in 1984 and 2019 Landsat images indicating retreat of 4300m, tributary separation and 5 km$^2$ lake expansion. A=Terminus tongue, B=Battle Glacier, G=Gilkey Glacier and T=Thiel Glacier. #3 is the location of an ice dammed lake that still forms and empties each year but is reduced in size. There had been two lakes near Point A that merged in 2019.

Specific Comments:

43: In fact large area change is dominated by large lakes, does this suggest you need a separate categorization for them?

45: I would avoid the ecological consequences discussion until later.  There is a complexity that is not examined here particularly in this region. For example the pro-glacial lakes in front of Excelsior Glacier as it has expanded has hosted numerous harbor seals, which will diminish as the glacier retreats form the lake.  Loso et al (2021) further examine the issue of river capture changes impact on fisheries.

50-65:  This section references numerous examples from the Himalayan and Peru Andes region.  The majority of references here should be specific to the study region Alaska. Glacier lake inventories from the Northern and Southern Patagonia icefield are probably more relevant as a comparison than the Himalayan or Peruvian Andes (Warren et al 2001; Loriaux and Casassa, 2013).  Is there value in indicating that the best analog regions varies depending on which range you are examining? Emmer et al (2016) notes that only 7.3% of the 8882 Cordillera Blanca lakes have an area greater than 0.1 km2, which is a different magnitude of size than the Alaska  glacier lakes.

150: Why are unconnected lakes used in this study, since they are neither fed by or connected to glaciers? I understand if it is just a validation tool, but then they should not be included in most of the analysis.

187:  Given the ephemeral nature, which will make an accurate inventory impossible, and there limited importance base on extent, is this study stronger with or without supraglacial lakes? I would suggest the data set is more robust without.  I offer this as a suggestion, but leave it to the authors to determine and will trust that answer.

213:  "…decreasing by 9 lakes", you indicate that 22 lakes were lost later in this section.

310:  This is where a reference to Brady Glacier would be useful given the number of ice dammed lakes, and that this a large low slope glacier (Capps et al, 2011; Pelto et al., 2013).

315: Neal (2007) is a good reference here as well for changing drainage from different ice dammed lakes of Tulsequah Glacier.

349:  Should note the trends observed both at Tulsequah Glacier  (Neal, 2007) and Bear Glacier (Wilcox et al 2014).

358: What is the importance of knowing normalized lake area variation by region?

387:  Yes this is true but here is where a closer look at the main area changing lakes will define the key topological elements.

388:  I do not see specific evidence supporting this assertion.  You have identified that supraglacial lakes are limited to a few large debris covered glaciers and have a small area. This does not suggest the range or role is expanding.

397: What are the similarities and just as importantly what are the differences?  Many of the debris covered glaciers in Alaska are not confined glacier tongues, but instead have an expanding terminus lobe either in a wide river valley (Battle, Donjek, Kaskawulsh, Lowell etc.) while others terminate on a coastal plain (Fingers, La Perrouse, Bering, Malaspina etc.).  In terms of debris cover change and proglacial lake changes how does this compare to the data from the Northern Patagoina Icefield from (Glasser et al 2016). "North Patagonian Icefield (NPI) in southern South America between 1987 and 2015 shows that the total amount of debris cover has increased over time, from 168 km$^2$ in 1987 to 307 km$^2$ in 2015. The area occupied by proglacial and ice-proximal lakes also increased from 112 to 198 km$^{2}$"

415:  I do not see the close correlation in the nature of the glaciers.  They differ in their climate setting, size, thickness, velocity, slope and elevation range.  The trend in lake dam type that is similar is relevant.  It must be emphasized that the size of the lakes between the two regions is vastly different.

435: No need to speculate given the glacier by glacier thinning data from Juneau Icefield and Stikine Icefield for example that illustrate variations Melkonian et al (2014 and 2016).

451:  Why use the Cordillera Blanca as an analog instead of the Northern Patagonia Icefield?

456: What do the observations from Tulsequah, Brady and Bear Glacier suggest as to diminishing number of ice dammed lakes?

**References**

Boyce, E., Motyka, R., and Truffer, M. Flotation and retreat of a lake-calving terminus, Mendenhall Glacier, southeast Alaska, USA. Journal of Glaciology, 53, 211–224, 2007.

Capps DM, Wiles GC, Clague JJ, Luckman BH. Tree-ring dating of the nineteenth-century advance of Brady Glacier and the evolution of two ice-marginal lakes, Alaska. The Holocene 21(4): 641–649. 2011.

Chen, F., Zhang, M., Guo, H., Allen, S., Kargel, J., Haritashya, U., and Watson, C. S.: Annual 30-meter Dataset for Glacial Lakes in High Mountain Asia from 2008 to 2017, Earth Syst. Sci. Data Discuss., 4275164, 1–29, https://doi.org/10.5194/essd-500 2020-57, 2021.

Clague, J., & Evans, S. Historic retreat of Grand Pacific and Melbern Glaciers Saint Elias Mountains, Canada: An analogue for decay of the Cordilleran ice sheet at the end of the Pleistocene? Journal of Glaciology, 39(133), 619-624. doi:10.3189/S0022143000016518, 1993.

Emmer, A. Klimeš, J. Mergili, M. Vilímek, V. Cochachin. A. 882 lakes of the Cordillera Blanca: an inventory, classification, evolution and assessment of susceptibility to outburst floods Catena, 147, 269-279, 2016.

Giffen B.A., Hall D.K., Chien J.Y.  Alaska: Glaciers of Kenai Fjords National Park and Katmai National Park and Preserve. In: Kargel J., Leonard G., Bishop M., Kääb A., Raup B. (eds) Global Land Ice Measurements from Space. Springer Praxis Books. Springer, Berlin, Heidelberg. https://doi.org/10.1007/978-3-540-79818-7_11, 2014.

Glasser, N. F., Holt, T. O., Evans, Z. D., Davies, B. J., Pelto, M., and Harrison, S. Recent Spatial and Temporal Variations in Debris Cover on Patagonian Glaciers. Geomorphology 273, 202–216, 2016.

Loriaux, T. and Casassa, G. Evolution of glacial lakes from the Northern Patagonia Icefield and terrestrial water storage in a sea-level rise context. Global and Planetary Change, 102, 33–40, 2013.

Loso, M. G., C. F. Larsen, B. S. Tober, M. Christoffersen, M. Fahnestock, J. W. Holt, and M. Truffer. 2021. Quo vadis, Alsek? Climate-driven glacier retreat may change the course of a major river outlet in southern Alaska. Geomorphology 384: 107701, 2021.

Melkonian, A. K., Willis, M. J., and Pritchard, M. E. Satellite-derived volume loss rates and glacier speeds for the Juneau Icefield, Alaska. *J. Glaciol.* 60, 743–760. doi: 10.3189/2014JoG13J181, 2014.

Melkonian, A. K., Willis, M. J., and Pritchard, M. E.: Stikine Icefield Mass Loss between 2000 and 2013/2014, Front. Earth Sci., 4, 89, https://doi.org/10.3389/feart.2016.00089, 2016.

Neal, E.G. Hydrology and glacier-lake-outburst floods (1987–2004) and water quality (1998–2003) of the Taku River near Juneau, Alaska: U.S. Geological Survey Scientific Investigations Report 2007–5027, 28 p, 2007.

Pelto, M., Capps, D., Clague, J.J. and Pelto, B.  Rising ELA and expanding proglacial lakes indicate impending rapid retreat of Brady Glacier, Alaska. Hydrol. Process., 27: 3075-3082. https://doi.org/10.1002/hyp.9913, 2013.

Pelto, M. Recent Climate Change impacts on Mountain Glaciers. Wiley-The Cryosphere Science Series, ISBN: 978-1-119-06811-2, 2017.

Trüssel, B., Motyka, R., Truffer, M., & Larsen, C. Rapid thinning of lake-calving Yakutat Glacier and the collapse of the Yakutat Icefield, southeast Alaska, USA. Journal of Glaciology, 59(213), 149-161. doi:10.3189/2013J0G12J081, 2013.

Warren, C., Benn, D., Winchester, V., & Harrison, S. Buoyancy-driven lacustrine calving, Glaciar Nef, Chilean Patagonia. Journal of Glaciology, 47(156), 135-146. doi:10.3189/172756501781832403, 2001.

Wilcox, A., Wade, A and Evans, E.  Drainage events from a glacier-dammed lake, Bear Glacier, Alaska: Remote sensing and field observations. Geomorphology 220: 41-49, 2014.

---

## Author Comment (AC3)

Dear Referee,

Thank you for taking the time to review our study and provide constructive feedback, suggestions which will improve the quality and understanding of the paper. Below we provide detailed responses to each of your comments, with our responses in blue.

This study provides insight into the evolution of ice-marginal lakes in the Alaska and NW Canada between 1984 and 2019 by employing supervised classification and semi-automated lake area delineation from Landsat images. The authors present novel findings and the text reads well. I find this study thematically suitable and of potential interest for the readers of The Cryosphere.

I have four more general (methodology-related) comments and a couple of specific ones.

The first general comment is related to inventory building. Using a semi-automated classification, you could possibly be missing some lakes identified as other features (false negatives); while you eliminated possible false positives by manual assignment of qualitative characteristics, this won't help you dealing with false negatives in a systematic way. Optimally, mapping outcomes of semi-automated classification would be checked against existing (e.g. sub-regional) inventory (you mentioned some in the intro), or manually prepared subset (e.g. manual mapping of 100 lakes to see the performance of semi-automated approach in terms of possible false negatives).

We agree that false negatives could be an issue within our dataset, though we minimize the likelihood by using more inclusive thresholds, then removing polygons manually. To add clarity in the manuscript, we will add a supplemental figure to illustrate what polygons look like after classification, after thresholding, and after manual cleaning to better explain the process. We also agree that checking outlines against an existing inventory would be useful, however, due to differences in methodology, criteria, and time period, comparison would introduce new uncertainties. Apart from the published study, we did compare our lakes to the dataset published by Field et al. (2021; https://doi.org/10.5194/tc-15-3255-2021), and found that 78 of the 85 manually delineated lakes were identified in our inventory. For the seven lakes not included, two are ice-dammed lakes (Summit Lake and Snow Lake) which are known to spend a significant amount of time drained or ice-filled, and therefore unidentifiable in our imagery. The other missed lakes are small, variable, marginal lakes which are likely missed due to differences in imagery dates (they are not visible in our mosaics). We will provide more explicit information and examples in the text about the types of lakes our inventory misses (i.e. drained, ice-filled). We decided not to include these types of lakes, even if we know their location, to avoid mixing methods and because although we know they exist, we can not provide an outline (since they don't appear in our Landsat mosaics) and therefore cannot assess change using our dataset. We aim to create a robust and repeatable dataset. Thus, our inventory presents a clear minimum in the number and location of lakes, with known false negatives which we will describe in the text.

The second comment is related to possibly missed fill-drain events (outbursts) typical for ice-dammed lakes (briefly mentioned on L104-107). This is actually quite important issue in my opinion (especially for formulating outburst hazard implications); I'm wondering whether any insight can be gained from histograms of pixel values used for mosaicking (for instance if two peaks of values in bare land and water spectra can indicate there was a lake outburst)? Please provide more discussion on this issue of possibly short-lived lakes (maybe a separate discussion section)

We agree that ice-dammed lakes that experience frequent drainage are very important for hazard implications, however, for this study, we focus on documenting decadal-scale patterns in lake area changes. Our dataset uses 5-year mosaics to detect a generalized outline for that period, which is unfortunately not sufficient to detect drainage events occurring on shorter timescales (often every 1-3

years). This is outside the scope of this study and will be addressed in a subsequent study. To further address this point, we will modify our introduction so that it is clear we are looking at decadal scale changes rather than rapid drainage events. As mentioned in our response above, we will also describe in greater detail why some lakes which are known to drain and fill are not included in our study.

The third one is related to dam type classification scheme. Your classes (Section 2.3.1) are defined in a clear, straightforward way. However, my experience from Peru is that I've been often facing cases where assignment to one of the classes was not at all straightforward in reality. For instance, I frequently observed lakes dammed by bedrock dam with discontinuous moraine cover (I ended up classifying these lakes as lakes as 'combined dams'). Sometimes, it was not possible to assign a dam to any of the classes, e.g. because of low quality / poor resolution of satellite imagery (and so I introduced 'not specified' dam category). Moreover, lake dam type can change in time (e.g. a shift from ice-dammed to bedrock-dammed is not rare). I'm also wondering whether you have observed any possibly landslide-dammed lakes in your inventory? Please comment on / discuss whether you've been facing similar issues when manually classifying lake dam types.

Thank you for pointing this out, as you have a unique appreciation for the difficulty in determining dam type. It is true that some dams are quite difficult to determine, and we classified them using our best judgement. We will add an additional paragraph within our discussion discussing the challenges in determining dam type, and emphasizing that our classification is our best interpretation, though it is limited by possible mixed dam types and poor imagery resolution. We will revise the explanation of Figure 2 as we wish to display typical lake behavior of each dam type, not necessarily provide an example of how we determined the dam type.

We do document a shift in dam type for some lakes, particularly from ice-dammed to bedrock-dammed (Figure R1.1). As each lake from each time step has its own polygon and dam type classification, this information is available within the dataset. However, for change over time, only one dam type can be classified for each individual lake, and therefore we use the most recent dam type for analysis. We do not observe any landslide-dammed lakes within our inventory. They likely do exist in Alaska, but our inventory is limited to lakes within 1 km of the RGI.

[Figure]

Figure R1.1. Example in shift from ice-dammed lake in 1984–1988 (A) to bedrock-dammed lake in 2016–2019 (B) for Terentiev Lake, which was dammed by the Columbia Glacier.

It is not fully clear how disappeared lakes (and there are many in fact) are considered and treated in statistics of total lake area change (e.g. Tab. 2), see also my specific comments; please provide more methodological details on that

Disappeared lakes are considered to have an area of zero km² after they disappear. We will include more explicit details in the revised manuscript about which lakes are included in which statistics (all inventoried lakes vs lakes only present in the most recent time period).

- - -

Specific comments:

L33: GLOF may also result from dam overtopping; dam breach is a sub-type (one of mechanisms) of dam failure in my understanding

Thank you-- we will reword to say "when a lake dam suddenly fails or is overtopped".

L56: can supraglacial lake also be located on debris-free glacier?

We only identified supraglacial lakes which are located within debris on debris-covered glaciers. We will be more explicit about this classification.

L85: you may consider confronting results of these previous Alaska-focusing studies in separate discussion section

We will add a discussion section focusing on Alaska-specific studies, both comparison to region-wide studies, as well as studies of a few specific lakes, as was also recommended by the second reviewer.

L94: A separate figure (workflow) depicting individual steps of the procedure, input and output data would be beneficial for readers

Thank you for this suggestion – we will add a workflow figure to clearly show what steps we used.

L112: can 'supraglacial debris' located on a glacier be distinguished from 'just debris' located elsewhere based on the spectral profile?

We did note spectral differences between supraglacial debris and 'just debris' (Figure R1.2), likely due to supraglacial debris generally being wetter and colder than off-glacier debris. Supraglacial debris often appeared as false positives after initial classification, as it had a spectral profile with more similarities to water, particularly in the NIR and SWIR bands (likely due to surficial melt associated with debris cover, dependent on debris thickness).

[Figure]

Figure R1.2. Examples of spectral profiles for water (dark blue), non-supraglacial debris (red), and supraglacial debris (light blue).

L117: pixel is areal unit (doesn't need to be squared)

This has been fixed – we were trying to describe a 7.5 by 7.5 pixel area, but we will just say an ~ 55 pixel area instead -- thank you.

L119: maybe you could specify date of images RGI is based on in your study area

The source imagery for the RGI in Region 01 is mostly from 2004-2010 (Kienholz et al., 2015). This information will be added to the description.

Fig. 2: These examples with false-color images are not very illustrative in terms of distinguishing different lake dam types; (e-f) instead of (d-f)

Thank you – here we aim to provide examples of typical lake behavior for each dam type, rather than examples of how we determined the dam type. We will fix our text to reflect this, and will correct the caption to (e-f).

Fig. 4: part a: you show drainages of lakes for individual periods (e.g. 11 drainages of moraine-dammed lakes in 1984-1988) – does it mean that you actually have insights into the within-period lake dynamics (and it is not blurred by mosaicking as described in methodology)? Please clarify

The example of 11 lakes in 1984-1988 is supposed to represent 11 lakes that are present in this time period which drain in a subsequent time period. We understand how this can be confusing and will add text to clarify that this does not represent within-period dynamics, but rather changes between periods.

L208: please unify Number of lakes (e.g. Fig. 4a) and frequency (e.g. Fig 5a) or explain the difference

These two represent the same thing (the number of lakes within each time period or bin), and the axes will be changed to reflect that.

L214: how did you actually deal with possibly merging lakes? Have you observed any such a case? Please comment

Lakes which merge or split at some point in time are given the same Lake ID, so that we do not get a false signal of a lake disappearing when the lakes merge or appearing when the lakes split. This happens most frequently for supraglacial lakes.

L224: 130 disappearing lakes from 791 total lakes is quite high number; if these were GLOFs, you observe 16.4 GLOFs per 100 lakes, which is extremely high ratio

The majority (103/130, 79%) of the lakes which disappeared are supraglacial lakes, many of which likely did not produce a GLOF. We will add a section in the discussion addressing the high number of disappearing likes (mainly due to the high variability of supraglacial lakes). Excluding supraglacial lakes, the new ratio would be 3.4 GLOFs (or rather drainage events, as we don't know how each lake drained) per 100 lakes.

L234-255: I suggest to start with % of lakes which actually experienced change and describe them in more detail in this chapter; taking into account lakes which did not experience areal change is confusing (and resulting in median change of 0.00 km$^2$ which is not very useful insight in my opinion)

Thank you for this suggestion. We will focus the results mainly on lakes which experienced change, and be explicit about which part of the dataset we are describing in each section.

Tab. 2: isn't this statistics biased when disappeared lakes are not considered – I mean, If you would consider 791 lakes instead of 661 lakes in this table, the overall pattern of lake area would be different I guess (count disappeared lakes as lake area decrease); further, I suggest to mention also min and max values, so the reader can get an idea about the range of observed values (median is ok, but I'm also interested in extremes); please consider re-designing this table

Thank you for this suggestion. As 103 of the 130 lakes which drained are supraglacial, the main change in statistics would be for the supraglacial lakes. Min and max do seem like valuable numbers to include and will be added to the table.

L257-258: you mentioned that most of the lakes did not experienced detectable change – how can then median change on subregional level when considering all lakes be 0.04-0.06 km$^2$ (I would expect 0.00 as well km$^2$)?

Thank you for catching this discrepancy. These are the median changes for lakes with detectable change. As lakes which experienced change are dominated by moraine-dammed lakes, these have the largest influence on the subregional area change. This will be clarified in the text.

Tab. 3: an interesting indicator could be lake area per deglaciated area

This is a very interesting suggestion. However, we are not aware of any existing dataset documenting deglaciated area within each subregion and producing such a dataset would be outside the scope of this study.

Fig. 7: please consider plotting relative cumulative lake area against relative cumulative lake count (that could provide clear insights what % of the largest lakes (count) represent what % of total area); analogically to Lorenz curve

Thank you for this suggestion. The second reviewer also suggested separating out the large lakes (>10 km$^2$) for a subanalysis to more clearly demonstrate how large lakes contribute the largest change in area. The plot that you suggest would be beneficial, although we think it would be easier to implement as a replacement to Figure 5. We feel that adding this type of plot for Figure 7 would be difficult since we are showing change over time, and have limited space on the map.

L281: some part of discussion are more results (e.g. section 4.3)

We will relocate some of the temporal trends discussion section to the results.

Fig. 8: please specify how many lakes are plotted in this figure

We will add an "n=" for each of the subplots.

L315: 'loss of an ice for'?

This reads "loss of an ice dam for", trying to explain that ice dam loss was attributed to ice thinning for 82% and 62% of land-terminating and lake-terminated glaciers, respectively.

Fig. 10: captions on x axis are confusing (if this is a change rate between two periods, both should be included, e.g. (1984-1988 to 1997-2001); (1997-2001 to 2007-2011); and (2007-2011 to 2016-2019), or similar)

We will make these changes to the x axis in Fig. 10 for clarity, as we are displaying the change rate between two periods.

L395-400: I think that important control of possible transferability of observed evolutionary patterns is topographical (relief) similarity (shape of a hypsometric curve of a mountain range); please consider taking this aspect into discussion

Thank you for this interesting suggestion. Transferability to other mountain ranges is an important aspect of the discussion, but as this would require the addition of new analyses and methods, we feel it is outside the scope of our current study.

L414: please replace 'basins' by 'parts of the study area'

This has been changed, thank you.

L459: second?

Yes, second, thank you for catching this mistake.

- - -

To sum up, I'm convinced this is valuable contribution to our understanding to dynamics of lake evolution in deglaciating mountain landscapes of Alaska and NW Canada. This study is undoubtedly worthy publishing as soon as some revisions are made. I suggest moderate to major revisions (especially methodological issues should be clarified, see my general comments).

Kind regards

Adam Emmer (Uni Graz, Austria)

Thank you again for your thorough and insightful review. We appreciate the time and effort you've invested to help improve our manuscript.

---

## Author Response (AR1)

Dear Referee 1,

Thank you for taking the time to review our study and provide constructive feedback, suggestions which improve the quality and understanding of the paper. Below we provide detailed responses to each of your comments, with our responses in blue. Specific text that has been added to our manuscript is in italics.

This study provides insight into the evolution of ice-marginal lakes in the Alaska and NW Canada between 1984 and 2019 by employing supervised classification and semi-automated lake area delineation from Landsat images. The authors present novel findings and the text reads well. I find this study thematically suitable and of potential interest for the readers of The Cryosphere.

I have four more general (methodology-related) comments and a couple of specific ones.

The first general comment is related to inventory building. Using a semi-automated classification, you could possibly be missing some lakes identified as other features (false negatives); while you eliminated possible false positives by manual assignment of qualitative characteristics, this won't help you dealing with false negatives in a systematic way. Optimally, mapping outcomes of semi-automated classification would be checked against existing (e.g. sub-regional) inventory (you mentioned some in the intro), or manually prepared subset (e.g. manual mapping of 100 lakes to see the performance of semi-automated approach in terms of possible false negatives).

We agree that false negatives could be an issue within our dataset, though we minimize the likelihood by using more inclusive thresholds, then removing polygons manually. To add clarity in the manuscript, we added a supplemental figure (Fig. S2) to illustrate what polygons look like after classification, after thresholding, and after manual cleaning to better explain the process. We also agree that checking outlines against an existing inventory would be useful, however, due to differences in methodology, criteria, and time period, comparison would introduce new uncertainties. Apart from the published study, we did compare our lakes to the dataset published by Field et al. (2021; https://doi.org/10.5194/tc-15-3255-2021), and found that 78 of the 85 manually delineated lakes were identified in our inventory. For the seven lakes not included, two are ice-dammed lakes (Summit Lake and Snow Lake) which are known to spend a significant amount of time drained or ice-filled, and therefore unidentifiable in our imagery. The other missed lakes are small, variable, marginal lakes which are likely missed due to differences in imagery dates (they are not visible in our mosaics).

We provide more explicit information and examples in the text about the types of lakes our inventory misses (i.e. drained, ice-filled; Section 4.7). We decided not to include these types of lakes, even if we know their location, to avoid mixing methods and because although we know they exist, we cannot provide an outline (since they don't appear in our Landsat mosaics) and therefore cannot assess change using our dataset. We aim to create a robust and repeatable dataset. Thus, our inventory presents a clear minimum in the number and location of lakes, with known false negatives which we describe in the text.

*L489-497: "Two primary challenges were encountered while creating the inventory, which should be understood before using this dataset or applying its findings. Due to the use of spectrally-based supervised classification for lake identification, only lakes that have the spectral characteristics of water appear within our inventory. This limitation results in the exclusion of some ice-dammed lakes which are either iceberg filled, or lake basins which appear dry in the five-year mosaics due to a lake that is more often drained than filled. Therefore, this inventory is a clear minimum of ice marginal lakes, with several known omitted lakes. Known lakes which were not identified in our inventory include Suicide Basin, Valdez Lake, Snow Lake, Lake Linda, and Summit Lake, all of which are ice-dammed. These lakes, which have experienced multiple drainage events (e.g., Jones and Wolken, 2019; Kienholz et al., 2020), are*

*important for individual GLOF hazard assessments, though they have little impact (0.4% of total lake area) on our assessment of regional decadal scale lake area trends by dam type."*

The second comment is related to possibly missed fill-drain events (outbursts) typical for ice-dammed lakes (briefly mentioned on L104-107). This is actually quite important issue in my opinion (especially for formulating outburst hazard implications); I'm wondering whether any insight can be gained from histograms of pixel values used for mosaicking (for instance if two peaks of values in bare land and water spectra can indicate there was a lake outburst)? Please provide more discussion on this issue of possibly short-lived lakes (maybe a separate discussion section)

We agree that ice-dammed lakes that experience frequent drainage are very important for hazard implications, however, for this study, we focus on documenting decadal-scale patterns in lake area changes. Our dataset uses 5-year mosaics to detect a generalized outline for that period, which is unfortunately not sufficient to detect drainage events occurring on shorter timescales (often every 1-3 years). This is outside the scope of this study and will be addressed in a subsequent study. To further address this point, we tried to make clear that we are looking at decadal scale changes rather than rapid drainage events. As mentioned in our response above, we also describe in greater detail why some lakes which are known to drain and fill are not included in our study (Section 4.7).

The third one is related to dam type classification scheme. Your classes (Section 2.3.1) are defined in a clear, straightforward way. However, my experience from Peru is that I've been often facing cases where assignment to one of the classes was not at all straightforward in reality. For instance, I frequently observed lakes dammed by bedrock dam with discontinuous moraine cover (I ended up classifying these lakes as lakes as 'combined dams'). Sometimes, it was not possible to assign a dam to any of the classes, e.g. because of low quality / poor resolution of satellite imagery (and so I introduced 'not specified' dam category). Moreover, lake dam type can change in time (e.g. a shift from ice-dammed to bedrock-dammed is not rare). I'm also wondering whether you have observed any possibly landslide-dammed lakes in your inventory? Please comment on / discuss whether you've been facing similar issues when manually classifying lake dam types.

Thank you for pointing this out, as you have a unique appreciation for the difficulty in determining dam type. It is true that some dams are quite difficult to determine, and we classified them using our best judgement. We added an additional paragraph within our "Inventory challenges" section (Section 4.7) discussing the challenges in determining dam type, and emphasizing that our classification is our best interpretation, though it is limited by possible mixed dam types and poor imagery resolution. We revised the explanation of Figure 3 (formerly Figure 2) as we wish to display typical lake behavior of each dam type, not necessarily provide an example of how we determined the dam type.

L498-504: *"Dam type classification was performed using manual visual interpretation for all lakes, with the most likely dam type identified in cases that are uncertain due to poor image resolution and/or possible mixed dam types (i.e. bedrock with moraine material; Emmer et al., 2017). A shift in dam type (e.g., ice dammed to bedrock dammed) was also documented for a few lakes, particularly where a lake becomes bedrock dammed after the loss of an ice dam (e.g., Terentiev Lake which was formerly dammed by Columbia Glacier). These lakes were classified by their most recent dam type when performing analysis on change over time. The small number of cases of transitioning dam types suggests this process is not particularly important for understanding the regional scale behavior of Alaska ice-marginal lakes discussed here."*

We do document a shift in dam type for some lakes, particularly from ice-dammed to bedrock-dammed (Figure R1.1). As each lake from each time step has its own polygon and dam type classification, this information is available within the dataset. However, for change over time, only one dam type can be classified for each individual lake, and therefore we use the most recent dam type for analysis. We do not observe any landslide-dammed lakes within our inventory. They likely do exist in Alaska, but our inventory is limited to lakes within 1 km of the RGI.

[Figure]

Figure R1.1. Example in shift from ice-dammed lake in 1984–1988 (A) to bedrock-dammed lake in 2016–2019 (B) for Terentiev Lake, which was dammed by the Columbia Glacier.

It is not fully clear how disappeared lakes (and there are many in fact) are considered and treated in statistics of total lake area change (e.g. Tab. 2), see also my specific comments; please provide more methodological details on that

Disappeared lakes are considered to have an area of zero km² after they disappear. We include this detail in the revised manuscript about which lakes are included in which statistics (all inventoried lakes vs lakes only present in the most recent time period).

- - -

Specific comments:

L33: GLOF may also result from dam overtopping; dam breach is a sub-type (one of mechanisms) of dam failure in my understanding

Thank you-- we reworded to say *"when a lake dam suddenly fails or is overtopped"*. (L34)

L56: can supraglacial lake also be located on debris-free glacier?

We only identified supraglacial lakes which are located within debris on debris-covered glaciers. We added text to say this explicitly on L148.

L85: you may consider confronting results of these previous Alaska-focusing studies in separate discussion section

We added text in both the introduction and discussion focusing on Alaska-specific studies, both comparison to region-wide studies, as well as studies of a few specific lakes, as was also recommended by the second reviewer.

L94: A separate figure (workflow) depicting individual steps of the procedure, input and output data would be beneficial for readers

Thank you for this suggestion – we added Figure 2 as a workflow figure.

[Figure]

**Figure 2: General workflow for creating the ice-marginal lake inventory for time periods 1984–1988, 1997–2001, 2007–2011, and 2016–2019.**

L112: can 'supraglacial debris' located on a glacier be distinguished from 'just debris' located elsewhere based on the spectral profile?

We did note spectral differences between supraglacial debris and 'just debris' (Figure R1.2), likely due to supraglacial debris generally being wetter and colder than off-glacier debris. Supraglacial debris often appeared as false positives after initial classification, as it had a spectral profile with more similarities to water, particularly in the NIR and SWIR bands (likely due to surficial melt associated with debris cover, dependent on debris thickness).

[Figure]

Figure R1.2. Examples of spectral profiles for water (dark blue), non-supraglacial debris (red), and supraglacial debris (light blue).

L117: pixel is areal unit (doesn't need to be squared)

This has been fixed – we were trying to describe a 7.5 by 7.5 pixel area, but we just say an ~ 55 pixel area instead -- thank you.

L119: maybe you could specify date of images RGI is based on in your study area

The source imagery for the RGI in Region 01 is mostly from 2004-2010 (Kienholz et al., 2015). This information has been added to the description in L102-103:

*"Imagery was limited to a 10 km buffer around the Randolph Glacier Inventory, with source imagery for glacier outlines mostly from 2004 to 2010 for Region 01 (RGI v6.0; RGI Consortium, 2017; Wang et al., 2012; Kienholz et al., 2015; Zhang et al., 2018)."*

Fig. 2: These examples with false-color images are not very illustrative in terms of distinguishing different lake dam types; (e-f) instead of (d-f)

Thank you – here we aim to provide examples of typical lake behavior for each dam type, rather than examples of how we determined the dam type. We revised our text to reflect this, and corrected the caption to (e-f):

Fig. 4: part a: you show drainages of lakes for individual periods (e.g. 11 drainages of moraine-dammed lakes in 1984-1988) – does it mean that you actually have insights into the within-period lake dynamics (and it is not blurred by mosaicking as described in methodology)? Please clarify

The example of 11 lakes in 1984-1988 is supposed to represent 11 lakes that are present in this time period which drain in a subsequent time period. We understand how this can be confusing and added text to clarify that this does not represent within-period dynamics, but rather changes between periods.

New Figure 5 caption: *"Number of lakes (a) and total area (b) of each dam type through time. The number of lakes in blue indicates the total number of lakes in that time period that were not present in 1984–1988; the number of lakes in red indicates the number of lakes present in that time period which drain in a subsequent time period. Light red indicates lakes which form after 1984–1988 but drain before 2016–2019. Bar widths correspond to imagery time intervals."*

L208: please unify Number of lakes (e.g. Fig. 4a) and frequency (e.g. Fig 5a) or explain the difference

*These two represent the same thing (the number of lakes within each time period or bin), and the axes were changed to reflect that.*

L214: how did you actually deal with possibly merging lakes? Have you observed any such a case? Please comment

*Lakes which merge or split at some point in time are given the same Lake ID, so that we do not get a false signal of a lake disappearing when the lakes merge or appearing when the lakes split. This happens most frequently for supraglacial lakes. We added text for clarification:*

*L136-137: "Lakes which coalesced or separated over time were given the same Lake ID to minimize misclassification of a lake forming or draining."*

L224: 130 disappearing lakes from 791 total lakes is quite high number; if these were GLOFs, you observe 16.4 GLOFs per 100 lakes, which is extremely high ratio

*The majority (97/124, 78% -- revised numbers) of the lakes which disappeared are supraglacial lakes, many of which likely did not produce a GLOF. Excluding supraglacial lakes, the new ratio would be 3.4 GLOFs (or rather drainage events, as we don't know how each lake drained) per 100 lakes. We added text for clarification (and had to adjust a few of the numbers):*

*L244-245: "Of the 124 lakes which drained, 97 (78%) were supraglacial, 20 (16%) were ice-dammed, and 7 (6%) were moraine-dammed."*

L234-255: I suggest to start with % of lakes which actually experienced change and describe them in more detail in this chapter; taking into account lakes which did not experience areal change is confusing (and resulting in median change of 0.00 km² which is not very useful insight in my opinion)

*Thank you for this suggestion. We revised to focus the results mainly on lakes which experienced detectable change, and tried to be explicit about which part of the dataset we are describing in each section.*

Tab. 2: isn't this statistics biased when disappeared lakes are not considered – I mean, If you would consider 791 lakes instead of 661 lakes in this table, the overall pattern of lake area would be different I guess (count disappeared lakes as lake area decrease); further, I suggest to mention also min and max values, so the reader can get an idea about the range of observed values (median is ok, but I'm also interested in extremes); please consider re-designing this table

*Thank you for this suggestion. As 103 of the 130 lakes which drained are supraglacial, the main change in statistics would be for the supraglacial lakes. Min and max are now included in Table 2 in the "Area range" column.*

L257-258: you mentioned that most of the lakes did not experienced detectable change – how can then median change on subregional level when considering all lakes be 0.04-0.06 km² (I would expect 0.00 as well km²)?

*Thank you for catching this discrepancy. These are the median changes for lakes with detectable change. As lakes which experienced change are dominated by moraine-dammed lakes, these have the largest influence on the subregional area change. This is clarified in the text L279.*

Tab. 3: an interesting indicator could be lake area per deglaciated area

This is a very interesting suggestion. However, we are not aware of any existing dataset documenting deglaciated area within each subregion and producing such a dataset would be outside the scope of this study.

Fig. 7: please consider plotting relative cumulative lake area against relative cumulative lake count (that could provide clear insights what % of the largest lakes (count) represent what % of total area); analogically to Lorenz curve

Thank you for this suggestion—we added relative cumulative lake count, area, and area change to a second panel in the new Figure 6, better representing that larger lakes contribute the most to total lake area and area change.

L281: some part of discussion are more results (e.g. section 4.3)

Thank you for this observation. We have decided to keep this section within the discussion, as we discuss the implications of the of the temporal trends and do not believe the small amount of results detracts from our discussion.

Fig. 8: please specify how many lakes are plotted in this figure

We added an "n=" for each of the subplots.

L315: 'loss of an ice for'?

This reads "loss of an ice dam for", trying to explain that ice dam loss was attributed to ice thinning for 82% and 62% of land-terminating and lake-terminated glaciers, respectively.

Fig. 10: captions on x axis are confusing (if this is a change rate between two periods, both should be included, e.g. (1984-1988 to 1997-2001); (1997-2001 to 2007-2011); and (2007-2011 to 2016-2019), or similar)

We made these changes to the x axis in new Figure 11 for clarity, as we are displaying the change rate between two periods.

L395-400: I think that important control of possible transferability of observed evolutionary patterns is topographical (relief) similarity (shape of a hypsometric curve of a mountain range); please consider taking this aspect into discussion

Thank you for this interesting suggestion. Transferability to other mountain ranges is an important aspect of the discussion, but as this would require the addition of new analyses and methods, we feel it is outside the scope of our current study.

L414: please replace 'basins' by 'parts of the study area'

This has been changed, thank you.

L459: second?

Yes, second, thank you for catching this mistake.

- - -

To sum up, I'm convinced this is valuable contribution to our understanding to dynamics of lake evolution in deglaciating mountain landscapes of Alaska and NW Canada. This study is undoubtedly worthy publishing as soon as some revisions are made. I suggest moderate to major revisions (especially methodological issues should be clarified, see my general comments).

Kind regards

Adam Emmer (Uni Graz, Austria)

Thank you again for your thorough and insightful review. We appreciate the time and effort you've invested to help improve our manuscript.
* * *
Dear Referee 2,

Thank you for taking the time to review our paper, provide constructive suggestions, and share your extensive knowledge to improve our study. Below we provide detailed responses to each of your comments, with our responses in blue. Specific text that has been added to our manuscript is in italics.

The authors have completed the first comprehensive inventory of Alaskan glacier marginal lakes, subdividing them both by dam type, topological relationship to the glacier, and size. This is a useful approach and as a baseline the number of lakes in each category and their respective area changes is useful. At present the paper does not provide the reader with sufficient context to understand the unique nature of many of the ice marginal lakes in Alaska, particularly the largest. There is a lack of referencing of previous studies that have explored specific areas identifying the relationship of the lakes and glaciers in away that would lend much better context to the inventory. An over reliance on references to the Himalayan and Peruvian Andes, which are not the best or in most cases even useful analogs underscores this issue. For Alaska it is indicated that large glacier lakes have an area greater than 10 km2, whereas inventories of both Cordillera Blanca and High Mountain Asia often use 0.1 km2, a two order of magnitude difference in scale (Emmer et al 2016; Chen et al 2021). The combination of these issues limits the value of the inventory data.

Thank you for this feedback. We acknowledge that we focused on comparisons to other regions, or to a few regional inventories, rather than including the many individual case studies to help describe the physical processes underlying the trends we observed in Alaska. We focus our introduction on regional lake inventories which classify dam type even if they are not the best analogue regions, to illustrate why dam type and topological position classification is important when analyzing regional lake trends. This type of study has not been done thus far in Alaska (nor other glaciated regions that might provide the most logical comparisons) and only occurs in limited regional-scale studies. We included an Alaska-specific paragraph in our revision of our introduction (L67-76) that describes some of the examples you provide to elaborate on the unique characteristics of the lakes found in this region.

There are several common features of the Alaskan lakes that are unusual leading to different behavior than for most glaciated alpine regions. I will review another of examples that illustrate this with referencing where appropriate. These illustrations are meant as examples, and not specific ones the authors may choose to use or need to specifically address.

Thank you for all these excellent examples and providing citations for the specific studies. As previously mentioned, we incorporated examples to illustrate Alaska-specific context.

An examination of Figure 5 illustrates how context is vital. The frequency of lakes is broken down by actual area in 2016-2019 in panel A, with the largest four groups greater than 2 km2 in area representing few of the total, but dominating the spatial area change noted in panel B. There are a significant number of Alaskan glacier lakes with an area greater than 20 km2. There are some specific unique characteristics of these larger lakes.

Most of the largest Alaskan glacier lakes are found in a coastal plain environment and are impounded by a coastal shoreline systems and/or proglacial deltas formed when the glacier terminated at the lake margin and outwash plains more than by moraine. They can be categorized as moraine dammed. But there is no potential for a dam failure to cause issues at near coastal lakes such at Mendenhall Glacier, Yakutat Glacier, Excelsior Glacier or Bear Glacier. All of these lakes are less than 20 m in elevation. Johnstone Lake the proglacial lake of Excelsior Glacier it is less than 1 km from the ocean with the highest elevation between tidewater and the lake being ~10 m (Figure 1). This lake has expanded from 9-18 km2 from 1994-2018, but still poses no GLOF risk. At Bear Glacier terminates in expanding Bear Glacier Lake,

which is 0.5 km from the ocean with a maximum elevation of ~7 m between the ocean and the lake. Malaspina Lake (~85 km2) is the proglacial lake at the southeast margin of Malaspina Glacier. It is 2 km from the ocean with a maximum elevation of about ~10 m. Grand Plateau Glacier Lake (~45 km2 Figure 2) is in a similar position.

*While it is true that these lakes do not pose any GLOF risk, we emphasize that GLOF risk is not the main motivating factor behind this inventory. We modified our introduction to better communicate that while GLOF hazards are important and will be the focus of a subsequent study, here we are interested in looking at decadal scale changes to glacial lakes in Alaska (see aims L85-89) to better understand where lakes are on the landscape, what changes are occurring, and attempting to parse whether these changes can be generally categorized by dam type and topological position to capture decadal scale trends. Therefore, we believe it is pertinent to include these large, low lying proglacial lakes in our inventory as they do have an impact on ecosystems, sediment transport, glacier dynamics, etc, even if they do not pose a GLOF risk.*

*We also add L110-111 for clarification: "Owing to this limitation, our focus here is on decadal changes in lake number and area, not sub-annual lake dynamics relevant for GLOF frequency."*

Many of these lakes have filled basins developed by the loss of the piedmont lobe terminus formed when the glacier exited the mountains onto low slope forelands/coastal zones. Giffen et al (2014) examined retreat and expanding lakes of glaciers in the Kenai and Katmai regions. ie. Bear, Fourpeaked, Spotted (Figure 3), Hallo Glaciers are examples of glacier lakes filling former piedmont lobe depressions. Mendenhall, Grand Plateau are examples from southeast Alaska. This is not a proglacial lake type seen in High Mountain Asia or the Andes north of Patagonia. Should you separate out this group of moraine dammed proglacial lakes terminating in a coastal environment? Loriaux and Casassa (2013) found a total lake area of the Northern Patagonia Icefield of 167.5± 8.4 km2 for 2011, an increase of 64.9% from 1945 (101.6±19.1 km2). They noted an 18 km2 expansion of San Quintin Lake accounting for 27% of the total expansion. This also is a coastal environment piedmont lobe terminus depression filled proglacial lake.

*We agree that it would be valuable to separate out these lakes (>10 km$^2$) for a subanalysis to show that the majority of area change documented was experienced by large lakes. We identify 19 lakes greater than 10 km$^2$ which account for 60% of the total area increase. Excluding these 19 lakes does not significantly change our results of median lake area change per dam type (see Fig. S6), and results in a total area increase of 41.5% rather than 59% from 1984–1988 to 2016–2019. Revised Figure 6 b (showing the relative cumulative sum of lake count, lake area, and area change) demonstrates that larger lakes contribute more to lake area and area change.*

The size of the lakes combined when depth also allows for the production of large tabular icebergs, such as seen at Excelsior, Ellsworth, Field and Yakutat in recent years. Trussel et al (2013) examined the rapid retreat of Yakutat Glacier in Harlequin Lake (in 2020 ~70 km2), with sections going afloat (Figure 4 and 5). They note this ability the ability to calve large tabular icebergs is largely due to the limited subaqueous melt compared to tidewater glaciers. The buoyancy also requires substantial water depth. The other unique element to these lakes given the large glaciers that feed them and the coastal setting, is they can be much deeper than examples from the Himalaya or Peruvian Andes. Trussel et al (2013) reported a depth of 325 m at the 2010 calving front of Yakutat Glacier. Loso et al (2021) note water depths of 330 m in Alsek Lake. For this inventory when do you draw the line between a glacial lake and one that is sufficiently tidally fed to not be a lake? Is Vitus Lake in front of Bering Glacier a part of the inventory?

*For this inventory, we use distance from the RGI (<1 km) to determine a glacial lake. Though a lake may have tidal influence, if there is a clear outline of the lake with a majority physical barrier (i.e. moraine with an outlet stream), we consider it an ice-marginal lake. We acknowledge that some of these low lying*

lakes may have tidal influence, however, their presence and change in area is linked to glacier dynamics, and therefore of relevance. We clarified our definition of a glacial lake in the text, and explain that some of the lakes may have a tidal influence. Vitus Lake is included within this inventory, and is classified as a proglacial moraine-dammed lake.

*L126-129: "For this study, any lake within 1 km of the RGI was considered an ice-marginal lake, acknowledging that some low-lying lakes (e.g., Vitus Lake) may have tidal influence. Lakes which are within 1 km of the RGI and unconnected from a glacial system are noted (Sect. 2.3.2) and treated separately (Table 2)."*

For Ice Dammed lakes it has to be emphasized that there are many that have had a long history of consistent GLOF events. Neal (2007) provides a detailed examination of glacier lake outbursts from beneath Tulsequah Glacier into the Taku River. They report on 41 GLOF's during the 1987-2004 period with the main source migrates from Tulsequah Lake to Lake No Lake, with little change in maximum outburst discharge. After 1984 a large proglacial lake has formed at the terminus of the glacier as well, which can help mitigate peak flows (Figure 6) (Pelto, 2017). Carrivick and Tweed (2016) list Tulsequah as the glacier lake with the most outburst floods. Wilcox et al (2014) report on the migration of Ice Lake which is ~17 km upglacier. This lake drains once or twice each year usually in late summer or early fall. This is another attribute worth noting that if a ice dammed lakes or supraglacial lakes drain into large proglacial lakes, there impact on discharge will be limited. Lake Linda on Lemon Creek Glacier is at the head of the glacier, and typically drains near mid-summer and does not refill.

While GLOF events, particularly from ice-dammed lakes, are an important factor in Alaska, we emphasize that documenting GLOF events is not the focus of this study. We are looking at decadal scale changes in lake area which unfortunately do not capture drainage events that occur on a sub-annual to annual timescale. This point was also raised by Reviewer 1, and we added a section to our discussion (Section 4.7 Inventory challenges) explicitly explaining why these types of lakes and events are difficult to capture within our study (i.e. five year mosaics, lakes which are frequently drained or ice-filled).

*L491-504: "Two primary challenges were encountered while creating the inventory, which should be understood before using this dataset or applying its findings. Due to the use of spectrally-based supervised classification for lake identification, only lakes that have the spectral characteristics of water appear within our inventory. This limitation results in the exclusion of some ice-dammed lakes which are either iceberg filled, or lake basins which appear dry in the five-year mosaics due to a lake that is more often drained than filled. Therefore, this inventory is a clear minimum of ice marginal lakes, with several known omitted lakes. Known lakes which were not identified in our inventory include Suicide Basin, Valdez Lake, Snow Lake, Lake Linda, and Summit Lake, all of which are ice-dammed. These lakes, which have experienced multiple drainage events (e.g., Jones and Wolken, 2019; Kienholz et al., 2020), are important for individual GLOF hazard assessments, though they have little impact (0.4% of total lake area) on our assessment of regional decadal scale lake area trends by dam type."*

Pelto et al (2013) examined the expansion of nine lakes at distributary termini of Brady Glacier that are a combination of ice-dammed and proglacial and expanded from 8.5 km2 in 1948 to 18.2 km2 in 2010 (Figure 7 and 8). How did you classify these lakes? Capps et al (2011) identified a depth of 200 m for Abyss Lake. Brady Glacier an exceptional case having so many proglacial calving terminus fronts.

[Figure]

Figure R2.1. Ice-marginal lake outlines around Brady Glacier. Lakes noted are: A = Abyss, B = Bearhole, D = Dixon, N = North Deception, O = Oscar, Sd = South Dixon, and T = Trick.

Figure R2.1 illustrates our derived lake outlines for the lakes adjacent to Brady Glacier. Our inventory did not identify Oscar lake (O) due to the ice-choked nature of the lake. Spur lake (Sp) is an example of an ice-dammed lake that has decreased in area. Trick Lake (T) is an example of a lake which separated into two lakes. Bearhole (B) is an example of an expanding ice-dammed lake.

Merging of lakes can also occur for these large lakes. Loso et al (2021) note that glacier thinning in the region can alter which lake a glacier feeds in this case Alsek Lake (~60 km2) and Grand Plateau Lake (Figure 3). Another future merging of lakes will be at Fingers Glacier. The merging of lakes has been observed at Melbern Glacier (Clague and Evans, 1993) Gilkey Glacier (Figure 9) and Llewellyn Glacier (Pelto, 2017).

We also observed the merging of lakes, though most frequently with supraglacial lakes. If two lakes merged at some point in time, we gave them the same Lake ID so as not to have a false signal of a lake disappearing when the two lakes merged. We added the following text for clarification:

L136-137: *"Lakes which coalesced or separated over time were given the same Lake ID to minimize misclassification of a lake forming or draining."*

Specific Comments:

43: In fact large area change is dominated by large lakes, does this suggest you need a separate categorization for them?

As previously mentioned, we did a sub analysis excluding the 19 lakes larger than 10 km$^2$, and found that our results in terms of median change and trends per dam type did not change significantly. We added Figure 6b to try and show this relationship of larger lakes dominating the total area change when considering all lakes together.

45: I would avoid the ecological consequences discussion until later. There is a complexity that is not examined here particularly in this region. For example the pro-glacial lakes in front of Excelsior Glacier as it has expanded has hosted numerous harbor seals, which will diminish as the glacier retreats form the lake. Loso et al (2021) further examine the issue of river capture changes impact on fisheries.

Here we are just acknowledging that lake changes have ecological implications rather than diving into much of a discussion of the specifics. We believe it is important to state that ecological consequences are a piece of the equation, though we do not explore it extensively.

50-65: This section references numerous examples from the Himalayan and Peru Andes region. The majority of references here should be specific to the study region Alaska. Glacier lake inventories from the Northern and Southern Patagonia icefield are probably more relevant as a comparison than the Himalayan or Peruvian Andes (Warren et al 2001; Loriaux and Casassa, 2013). Is there value in indicating that the best analog regions varies depending on which range you are examining? Emmer et al (2016) notes that only 7.3% of the 8882 Cordillera Blanca lakes have an area greater than 0.1 km2, which is a different magnitude of size than the Alaska glacier lakes.

Thank you for this feedback. We now include a highlight of Alaskan studies in the text. It is true that the best analogue region depends on which range we are comparing it to, and we added clarification to our discussion Section 4.4 comparing various regions. We reference the studies in the Himalaya and Peruvian Andes because they are examples of studies which classify dam type and topological position, and demonstrate the importance of such classifications. We also believe it is important to put our work in its global context, in addition to an Alaska-centered lens.

150: Why are unconnected lakes used in this study, since they are neither fed by or connected to glaciers? I understand if it is just a validation tool, but then they should not be included in most of the analysis.

The criteria for our study is lakes within 1 km of the RGI. Therefore we include unconnected lakes, but specify that they are unconnected. Table 2 separates each dam type by location, showing how many lakes of each dam type were classified as unconnected. Apart from 4 moraine dammed lakes, all unconnected lakes are classified as bedrock dammed. Therefore, excluding unconnected lakes would only impact the statistics for bedrock dammed lakes, which experienced a median change of 0.0 km$^2$ whether or not unconnected lakes are included. In addition, comparable studies to ours also include unconnected lakes, such as Nie et al. (2017), Rounce et al. (2017), and Zhang et al. (2018), as they also use a distanced-base criteria for ice-marginal lakes.

187: Given the ephemeral nature, which will make an accurate inventory impossible, and there limited importance base on extent, is this study stronger with or without supraglacial lakes? I would suggest the data set is more robust without. I offer this as a suggestion, but leave it to the authors to determine and will trust that answer.

Supraglacial lakes have large spatial and temporal variations, and tend to be quite dynamic. Though this does make an inventory difficult, we think it is valuable to include these snapshots within the study as

supraglacial lakes contribute to glacier hydrology. As the lakes are classified by dam type and topological location, for most analysis we separate out supraglacial lakes and therefore the inclusion or exclusion of these lakes does not substantially change analysis or interpretation. By including differing lake types, different readers can focus on the information that is most relevant to them. For example, a researcher interested in debris-covered glacier mass loss may be particularly interested in supraglacial lakes because they play a role as "melt hotspots". Removing supraglacial lakes from analyses means that our study will not address this researcher's interest, with little downside produced by their inclusion.

213: "…decreasing by 9 lakes", you indicate that 22 lakes were lost later in this section.

Ice dammed lakes decreased by 6 lakes (we will correct this discrepancy) in total number (from 68 in 1984-1988 to 62 in 2016-2019). Twenty individual lakes were lost, meaning that 14 lakes appeared as well, to equal a decrease in 6 lakes overall. We clarified the text (L230-232) to better explain what each of these numbers represent.

310: This is where a reference to Brady Glacier would be useful given the number of ice dammed lakes, and that this a large low slope glacier (Capps et al, 2011; Pelto et al., 2013).

Thank you -- the addition of specific examples strengthens our discussion. We include a reference to Brady Glacier and the associated references.

L332-333: *"Ice-dammed lakes are primarily located next to clean-ice glaciers (65%), with larger, lower slope areas and positive hypsometric indices (bottom-heavy; Fig. 10), such as are found adjacent to Brady Glacier (Capps et al., 2011; Pelto et al., 2013)."*

315: Neal (2007) is a good reference here as well for changing drainage from different ice dammed lakes of Tulsequah Glacier.

L337-338: *"The formation of new conduits alongside or beneath the glacier could also be influencing ice-dammed lake drainage (e.g., Tulsequah Glacier; Neal, 2007; Post and Mayo, 1971)."*

349: Should note the trends observed both at Tulsequah Glacier (Neal, 2007) and Bear Glacier (Wilcox et al 2014).

We incorporated these references -- thank you.

L369-372: *"The observed decrease in number and area of ice-dammed lakes in Alaska suggests an overall decrease in GLOF hazards from these types of lakes, although the specifics of remaining lakes need to be evaluated individually (e.g., Suicide Basin, Kienholz et al., 2020; Tulsequh Glacier, Neal, 2007; Bear Glacier, Wilcox et al., 2014)."*

358: What is the importance of knowing normalized lake area variation by region?

We present normalized lake area in order to account for differences in glacierized area between the different subregions and therefore make inter-region comparisons more comparable. We wanted to look at the normalized lake area for each region to see whether the amount of lake area per region is consistent given the large variation in glacierized area, or rather, whether lake area scales with glacierized area.

387: Yes this is true but here is where a closer look at the main area changing lakes will define the key topological elements.

We reanalyzed the data excluding lakes >10 km$^2$, and found that moraine-dammed lakes still had the largest median change and rate of change.

388: I do not see specific evidence supporting this assertion. You have identified that supraglacial lakes are limited to a few large debris covered glaciers and have a small area. This does not suggest the range or role is expanding.

Here we are looking at the increasing median rate of change for supraglacial lakes and decreasing median rate of change for moraine-dammed lakes (Figure 11a), such that the median rate of change is nearly the same for supraglacial and moraine-dammed lakes between the time periods of 2007-2011 and 2016-2019. We believe this is significant, and try to use qualifying language such as "could be". We believe the median rates of change supports an increase in the role of supraglacial lakes, not necessarily that the entire mechanism for change is shifting. We clarified the language to better present this idea.

L415-417: *"Together, these data suggest the ice-marginal lakes of Alaska and northwestern Canada could be undergoing a transition, with an increase in the role of supraglacial lakes in lake area expansion."*

397: What are the similarities and just as importantly what are the differences? Many of the debris covered glaciers in Alaska are not confined glacier tongues, but instead have an expanding terminus lobe either in a wide river valley (Battle, Donjek, Kaskawulsh, Lowell etc.) while others terminate on a coastal plain (Fingers, La Perrouse, Bering, Malaspina etc.). In terms of debris cover change and proglacial lake changes how does this compare to the data from the Northern Patagoina Icefield from (Glasser et al 2016). "North Patagonian Icefield (NPI) in southern South America between 1987 and 2015 shows that the total amount of debris cover has increased over time, from 168 km2 in 1987 to 307 km2 in 2015. The area occupied by proglacial and ice-proximal lakes also increased from 112 to 198 km2"

We have focused our comparisons on other studies that similarly split their analysis of lake behavior on dam type and lake location. We acknowledge that the style of glaciation in NPI (and other glaciated regions of the world) might provide other suitable comparisons, but in the absence of similar comprehensive yet also detailed glacial lake studies, we feel that the former comparisons are most suitable for the scope of this paper. For example, Glasser et al. (2016) found an overall increase in lake area, though they do not separate areas for proglacial and ice-dammed lakes which makes comparison difficult. In our revisions, we note that the best analog region depends on which Alaska subregion is considered.

L421-423: *"Alaska's glaciers cover a large spatial extent, spanning different climatic, geologic, and glacial environments (Section 1). This makes the region particularly unique in relation to other glacierized regions, though the best analogue region depends on which Alaskan mountain range is considered."*

415: I do not see the close correlation in the nature of the glaciers. They differ in their climate setting, size, thickness, velocity, slope and elevation range. The trend in lake dam type that is similar is relevant. It must be emphasized that the size of the lakes between the two regions is vastly different.

Thank you-- we are most interested in the documented change in dam type and are using the Cordillera Blanca as an example of shifting dam types. We are comparing to studies which classify lakes by dam type, whereas most other lake inventories do not provide this additional classification.

435: No need to speculate given the glacier by glacier thinning data from Juneau Icefield and Stikine Icefield for example that illustrate variations Melkonian et al (2014 and 2016).

Thank you -- we believe speculative language is still appropriate here as we are referring to future events, but added additional information:

L464-467: *"We therefore expect that as glaciers continue to retreat in Alaska, proglacial lake presence, formation, or detachment will influence glacier mass balance, increasing thinning and glacier velocity where lakes are present, though variations in thinning have been observed (e.g., the Juneau and Stikine icefields; Melkonian et al., 2014, 2016)"*.

451: Why use the Cordillera Blanca as an analog instead of the Northern Patagonia Icefield?

We agree with your suggestion that the Northern Patagonia Icefield makes an excellent analogue for current conditions, however, we chose to use the Cordillera Blanca to illustrate what could occur farther into the future. Again, we focus on studies which document lake dam type.

456: What do the observations from Tulsequah, Brady and Bear Glacier suggest as to diminishing number of ice dammed lakes?

The inventory revealed variable (and interesting!) behavior across the ice dammed lakes we studied. We find that these three glacier examples contain persistent ice-dammed lakes over the study period, but when we assess all ice-dammed lakes in the region, we find that this class of lakes decreased in area and number.  We acknowledge that the variability in individual lake behavior provides an interesting topic for future studies, but the scope of the current study limits the level of detail we can include on these individual lakes. We include Suicide Basin as an example of an ice-dammed lake that has formed in the recent past (2011), and could illustrate what kind of ice-dammed lakes will form in the future, i.e. an example of a tributary valley where ice has retreated and created a lake basin. The question is how these competing processes will play out in terms of changing total number of ice-dammed lakes, ice thinning and retreating, and removing ice dams vs tributary ice retreat and the formation of lake basins. This is where modeling of glacier retreat would be useful, though it is outside the scope of this study.

Thank you again for your excellent suggestions that improved our manuscript, and for including references which were helpful while revising the manuscript.  We appreciate the time and effort you've invested and the wealth of specific knowledge you have shared.

---

## Author Response (AR2)

Dear Dr. Homa Kheyrollah Pour,

Thank you for your comments and for handling our manuscript. We have addressed your suggested and essential revisions below, with our responses in blue.

**Suggested Revisions:**

To add new discussion Section 4.7 and clarifications of some methodological aspects.

Our interpretation of this comment from Reviewer #1 was that they were acknowledging the changes we made rather than suggesting new revisions, as they said, "I appreciate new discussion Section 4.7 and clarifications of some methodological aspects".

**Suggested Revisions:**

195: The authors make the case here that supraglacial lakes are not as important in terms of area or change. I suggest discussing supraglacial lakes last of the main types in section 3.0 and 3.3.

Thank you for this suggestion. We considered this comment, but feel that the current structure provides the best presentation of our results.

235: The authors note, "There are 19 lakes (17 moraine-dammed, one bedrock-dammed, and one ice-dammed) with individual areas greater than 10 km$^2$, accounting for 288.7 km$^2$ (60%) of the total area growth from 1984–1988 to 2016–2019." These 19 lakes hence are the most critical to lake change area in Alaska. I would suggest providing a table indicating the specific characteristics of these 19 lakes, that may indicate shared characteristics. Please include elevation and if the lakes are coastal environment piedmont lobe terminus depression filled proglacial lakes.

We have added a table as suggested to our supplementary material (Table S1). In addition, we added relevant text summarizing this table:

*L237-240: "Of these 19 lakes, 14 occupy piedmont lobe depressions (defined here as a basin which is not constrained by valley walls), and about half are found within a coastal plain environment, with four lakes located within 2 m of sea level."*

**Essential revisions:**

Figure 6. This is an accurate figure, but visually does not convey the information in an easily identifiable fashion.

We have revised this figure by removing text, and improved clarity by adding a legend and a grid to help with interpretation.

Figure 9 X-axis needs additional unit labeling for accurate interpretation.

We added additional unit labels to aid in accurate interpretation.

Thank you again for all the helpful suggestions that have improved our manuscript.

Sincerely,

Brianna Rick, on behalf of the co-authors